# SIRT5 is a proviral factor that interacts with SARS-CoV-2 Nsp14 protein

**Marius Walter**[1¤]*, **Irene P. Chen**[2,3,4], **Albert Vallejo-Gracia**[2,3,4], **Ik-Jung Kim**[1],
**Olga Bielska**[1], **Victor L. Lam**[3], **Jennifer M. Hayashi**[2,3,4], **Andrew Cruz**[1], **Samah Shah**[1], **Frank W. Soveg**[2,3,4], **John D. Gross**[3,5], **Nevan J. Krogan**[2,3,4,5], **Keith R. Jerome**[6,7],
**Birgit Schilling**[1], **Melanie Ott**[2,3,4,8], **Eric Verdin**[1]*

**1** Buck Institute for Research on Aging, Novato, California, United States of America, **2** Gladstone Institutes, San Francisco, California, United States of America, **3** University of California San Francisco, San Francisco, California, United States of America, **4** QBI COVID-19 Research Group (QCRG), San Francisco, California, United States of America, **5** Quantitative Biosciences Institute (QBI), University of California San Francisco, San Francisco, California, United States of America, **6** Vaccine and Infectious Disease Division, Fred Hutch Cancer Center, Seattle, Washington, United States of America, **7** Department of Laboratory Medicine and Pathology, University of Washington, Seattle, Washington, United States of America, **8** Chan Zuckerberg Biohub, San Francisco, California, United States of America

¤ Current address: Vaccine and Infectious Disease Division, Fred Hutch Cancer Center, Seattle, Washington, United States of America
* mwalter2@fredhutch.org (MW); everdin@buckinstitute.org (EV)

**Data Availability Statement:** The data supporting the findings of this study are available within the paper and its Supplementary files. RNA-seq data have been deposited to the GEO repository (GSE188382). Code developed for RNA-seq

## Abstract

SARS-CoV-2 non-structural protein Nsp14 is a highly conserved enzyme necessary for viral replication. Nsp14 forms a stable complex with non-structural protein Nsp10 and exhibits exoribonuclease and N7-methyltransferase activities. Protein-interactome studies identified human sirtuin 5 (SIRT5) as a putative binding partner of Nsp14. SIRT5 is an NAD-dependent protein deacylase critical for cellular metabolism that removes succinyl and malonyl groups from lysine residues. Here we investigated the nature of this interaction and the role of SIRT5 during SARS-CoV-2 infection. We showed that SIRT5 interacts with Nsp14, but not with Nsp10, suggesting that SIRT5 and Nsp10 are parts of separate complexes. We found that SIRT5 catalytic domain is necessary for the interaction with Nsp14, but that Nsp14 does not appear to be directly deacylated by SIRT5. Furthermore, knock-out of SIRT5 or treatment with specific SIRT5 inhibitors reduced SARS-CoV-2 viral levels in cell-culture experiments. SIRT5 knock-out cells expressed higher basal levels of innate immunity markers and mounted a stronger antiviral response, independently of the Mitochondrial Antiviral Signaling Protein MAVS. Our results indicate that SIRT5 is a proviral factor necessary for efficient viral replication, which opens novel avenues for therapeutic interventions.

## Author summary

SARS-CoV-2 is a pathogen of global concern. After cellular entry, SARS-CoV-2 hijacks the cellular machinery, and the viral proteins physically interact with hundreds of human proteins. Here we described the interaction between SARS-CoV-2 protein Nsp14, a key enzyme necessary for viral replication, and human sirtuin 5 (SIRT5), a protein deacylase

analysis is available on Github (github.com/mariuswalter/SIRT5_paper). Mass spectrometric raw data have been deposited to the MassIVE repository (MSV000088589) and are also available at ProteomeXchange (PXD030530).

**Funding:** This study was funded through institutional support from the Buck Institute for Research on Aging. IPC received support from the NIH (F31 AI164671-01). M.O. gratefully acknowledges support through gifts by Pamela and Edward Taft, and the Roddenberry Foundation. We acknowledge the support of instrumentation from the NCRR shared instrumentation grant 1S10 OD016281 (Buck Institute) for Mass Spectrometry Analysis, and the NIH Shared Instrumentation Grant 1S10OD010786-01 (UC Davis) for RNA-sequencing. The funders had no role in study design, data collection and analysis, decision to publish, or preparation of the manuscript.

**Competing interests:** The Krogan Laboratory has received research support from Vir Biotechnology, F. Hoffmann-La Roche, and Rezo Therapeutics. Nevan J. Krogan has financially compensated consulting agreements with the Icahn School of Medicine at Mount Sinai, New York, Maze Therapeutics, Interline Therapeutics, Rezo Therapeutics, GEn1E Lifesciences, Inc. and Twist Bioscience Corp. He is on the Board of Directors of Rezo Therapeutics and is a shareholder in Tenaya Therapeutics, Maze Therapeutics, Rezo Therapeutics, and Interline Therapeutics.

that removes succinyl and malonyl groups from lysine residues. We showed that SIRT5 strongly interacts with Nsp14 and that SIRT5 catalytic domain is necessary for the interaction, despite Nsp14 not being directly deacylated by SIRT5. Furthermore, we found that knocking out or inhibiting SIRT5 reduced SARS-CoV-2 viral levels in cell-culture experiments, and that SIRT5 knock-out cells mounted a stronger antiviral response. Altogether, our result indicates that SIRT5 is a proviral factor necessary for efficient viral replication. SIRT5 is a critical metabolic enzyme that regulates several important metabolic processes, but its role during disease and infection is currently unknown. Our work suggests that SIRT5, and potentially other sirtuins, could act as a bridge between cellular metabolism and the innate immune responses against viral infections.

## Introduction

Severe acute respiratory syndrome coronavirus 2 (SARS-CoV-2) is a pathogen of global concern that needs no further introduction. After cellular entry, SARS-CoV-2 hijacks the cellular machinery, and the viral proteins physically interact with hundreds of human proteins [1–4]. In most cases, however, the exact nature of the interactions and their functions and relevance during viral infection remain unknown.

SARS-CoV-2 encodes two large open reading frames, ORF1a and ORF1b, that are processed into 16 non-structural proteins after proteolytic cleavage by viral proteases. The 16 non-structural proteins, Nsp1 to Nsp16, are involved in every aspect of viral replication and are highly conserved in coronaviruses. Coronavirus Nsp14 protein is part of the replication-transcription complex and has two conserved domains with distinct functions. The N-terminal domain acts as a 3' to 5' exoribonuclease (ExoN), and the C-terminal domain displays RNA cap guanine N7-methyltransferase (MTase) activity (Fig 1A) [5–8]. The N-terminal ExoN domain provides proofreading activity during RNA replication, allowing the removal of mismatched nucleotides introduced by the viral RNA polymerase [9–11]. This proofreading activity ensures a high level of fidelity during RNA replication and is unique among RNA viruses [12,13]. Coronaviruses and related viruses in the order nidovirales have some of the largest genomes (26–32 kb) among known RNA viruses [14], and the acquisition of ExoN activity is thought to have allowed nidoviruses to evolve these large genomes [9,15]. The C-terminal MTase domain of Nsp14 is an S-adenosyl methionine (SAM)-dependent methyltransferase critical for viral RNA capping that methylates the 5' guanine of the Gppp-RNA cap at the N7 position [6,7]. The 5' cap is important for viral mRNA stability and translation and for escaping host innate antiviral responses. Importantly, Nsp14 forms a stable complex with the non-structural protein Nsp10, a small zinc-binding co-factor with no reported enzymatic activity on its own [7,10]. Nsp10 binds and stabilizes the N-terminal ExoN domain of Nsp14 and is necessary for ExoN activity, but not for MTase activity. Interestingly, mutations that abolish ExoN activity cause a lethal phenotype in SARS-CoV-2 and MERS-CoV, but not in SARS-CoV or other coronaviruses [16], suggesting that ExoN has additional functions beyond its proofreading activity. Indeed, Nsp14 triggers translational shutdown, participates in evasion of innate immunity, activates proinflammatory signals, and mediates viral recombination [17–20].

Large-scale protein-protein interaction analyses of SARS-CoV-2 and human proteins revealed putative interacting partners for all of the SARS-CoV-2 proteins. Several independent studies, from us and others, suggested that SARS-CoV-2 Nsp14 protein interacts with human sirtuin 5 (SIRT5) [1–4]. Sirtuins are a family of conserved protein deacylases and mono-ADP-

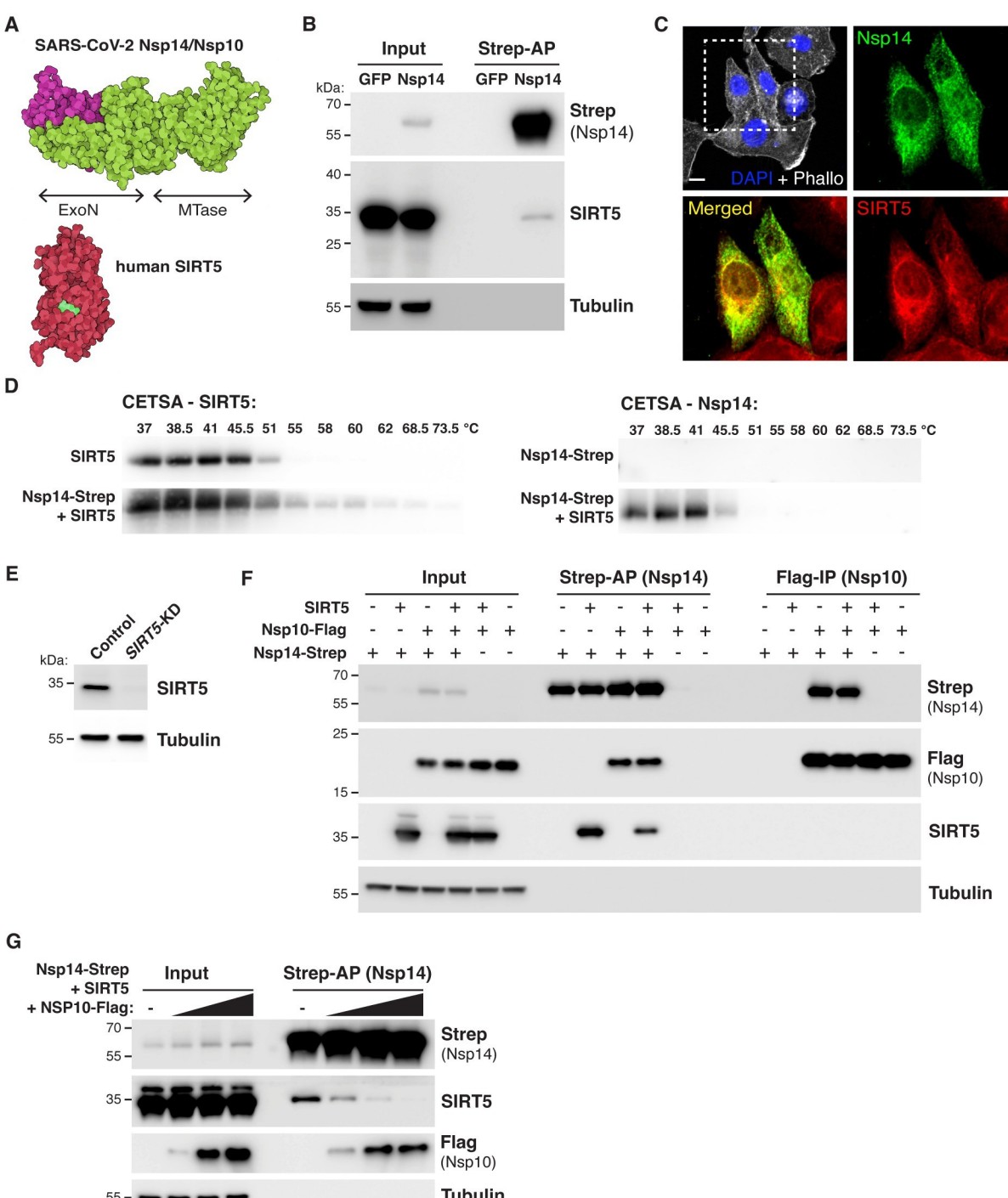

**Fig 1. SARS-CoV-2 Nsp14 interacts with human SIRT5. A.** Cartoon representation of the protein structure of Nsp14/Nsp10 (PDB 7N0B) and SIRT5 (PDB 3YIR) shows the Nsp14 N-terminal ExoN domain and C-terminal MTase domain. **B.** Affinity-purification of Nsp14-strep and co-purification of endogenous SIRT5 after transfection in HEK293T cells, as shown by western blot. **C.** Immunofluorescence of transfected Nsp14-Strep and endogenous SIRT5 in A549 cells. **D.** CETSA in HEK293T cells transfected with Nsp14-step and/or SIRT5, showing an increase in the stability of SIRT5 and Nsp14 by western blot. **E.** Western blot showing the absence of SIRT5 in *SIRT5*-KD HEK293T cells. **F.** Strep-tag affinity-purification or Flag-tag immunoprecipitation, followed by western blot, after transfection with Nsp14-strep, Nsp10-flag and SIRT5 expression constructs. SIRT5 does not interact with Nsp10. 0.5 μg of each construct or of empty control plasmids were transfected in *SIRT5*-KD HEK293T cells in a six-well plate. **G.** Strep-tag affinity-purification and western blot after transfection of Nsp14-strep, SIRT5 and increasing concentrations of Nsp10-tag indicate competitive binding of SIRT5 and Nsp10. 0.5 μg of Nsp14-strep and SIRT5 plasmid were used in a 6-well plate, with 0, 0.5, 1 or 2 μg of Nsp10-Flag.

ribosyltransferases found in organisms ranging from bacteria to humans. Sirtuins use nicotinamide adenine dinucleotide (NAD) as a co-substrate and are important regulators of cellular metabolism and aging [21,22]. Most sirtuins act as NAD-dependent protein deacetylases, removing acetyl groups from lysine residues and, as such, tightly connect post-translational protein regulation with cellular metabolism. The seven mammalian sirtuins (SIRT1–7) are found in different cellular compartments. They deacylate histones and transcriptional regulators in the nucleus and also specific proteins in the cytoplasm and mitochondria. Sirtuins are crucial regulators of cellular metabolism and energy homeostasis and have emerged as key regulators of aging and age-related diseases.

SIRT5 is unique among the seven mammalian sirtuins. It is only a weak protein deacetylase, but it efficiently removes longer-chain acyl groups from proteins, such as succinyl, malonyl or glutaryl groups [23,24]. By preferentially catalyzing the removal of these negatively charged acidic modifications, SIRT5 functions as the main cellular desuccinylase, demalonylase, and deglutarylase [24–26]. SIRT5 is predominantly found in the mitochondria, but also exerts regulatory activity in the cytoplasm. It is involved in several important metabolic processes, such as glycolysis, fatty acid oxidation and ketone body production [27]. Despite elevated succinylation or malonylation levels in several tissues, no obvious phenotype or abnormalities are observed in *Sirt5* knockout mice under basal conditions [28]. The roles of SIRT5 in disease, infection, and aging, are unclear.

Here we investigated the role of SIRT5 during infection with SARS-CoV-2. We showed that SIRT5 interacts with Nsp14, but not with its cofactor Nsp10, and that SIRT5 catalytic activity is necessary for the interaction. Furthermore, knock-out or inhibition of SIRT5 reduced viral levels in cell-culture experiments, revealing that SIRT5 is a proviral factor necessary for efficient viral replication.

## Results

### SARS-CoV-2 Nsp14 interacts with human SIRT5

Protein-protein interaction mass-spectrometry studies suggested that SARS-CoV-2 Nsp14 binds to SIRT5 [1–4]. We first sought to confirm and characterize the nature of this interaction. We used a mammalian expression vector developed by Gordon et al. that contains Nsp14 with a 2xStrep affinity tag (Nsp14-strep) that can be used for affinity purification [1]. Plasmids expressing Nsp14-strep or a GFP control were transfected into HEK-293T cells for 48 hours, and tagged proteins were purified by affinity purification using magnetic beads. Using western blots, we found that SIRT5 was specifically co-purifying with Nsp14, confirming published mass spectrometry results (Fig 1B). Immunofluorescence in human alveolar basal epithelial A549 cells transfected with Nsp14 expression plasmid further showed that Nsp14 and SIRT5 co-localized into the same cellular compartments, with a predominantly cytoplasmic and perinuclear localization (Fig 1C).

We next used Cellular Thermal Shift Assay (CETSA) to quantify the changes in the thermal stability of Nsp14 and SIRT5 in intact cells. The thermal stability of proteins changes upon ligand binding, and CETSA can be used to record the strength of the interaction in the physiological context. HEK-293T cells were transfected with plasmids expressing Nsp14 and SIRT5, either alone or in combination, and the shift in thermal stability was assessed by western blot (Fig 1D). We observed an important increase in the stability of both proteins when they were co-transfected together. Nsp14, in particular, was poorly expressed and barely detectable when transfected alone, but a strong signal appeared in the presence of SIRT5. Overall, these initial observations showed that SIRT5 and Nsp14 were interacting in human cells and that SIRT5 strongly stabilized Nsp14 expression.

Nsp14 forms a stable complex with the small viral protein Nsp10 [7,10]. We thus tested whether SIRT5 also interacted with Nsp10, or whether they were parts of independent complexes. To eliminate endogenous expression of SIRT5, we generated a *SIRT5* knockdown cell line (*SIRT5*-KD), using CRISPR interference in HEK-293T cells [29]. Several guide RNAs were tested, and one was selected for the rest of the study. SIRT5 was undetectable in this cell line (Fig 1E). Expression plasmids for SIRT5, Nsp14-strep and Nsp10 with a Flag tag (Nsp10-Flag) were transfected alone or in combination for 48 hours in *SIRT5*-KD cells. Proteins were then co-purified either by strep affinity purification (Strep-AP) or Flag immunoprecipitation (Flag-IP) (Fig 1F). Strep-AP confirmed that both SIRT5 and Nsp10 interacted with Nsp14. By contrast, pulling down Nsp10 by Flag-IP showed that only Nsp14 co-purified with Nsp10. This indicates that Nsp10 and SIRT5 do not interact. Besides, the SIRT5 signal after Strep-AP appeared to be lower in the presence of Nsp10, and we hypothesized that Nsp10 and SIRT5 compete for Nsp14 binding. To test this hypothesis, Nsp14, SIRT5, and an increasing quantity of Nsp10 plasmids were co-transfected in *SIRT5*-KD cells. SIRT5 binding was lost with high concentrations of Nsp10 (Fig 1G). Thus, SIRT5 and Nsp10 competitively bind Nsp14, and Nsp14/SIRT5 and Nsp14/Nsp10 likely form independent complexes.

## SIRT5 catalytic activity is necessary for the interaction with Nsp14

SIRT5 is the main cellular desuccinylase, demalonylase, and deglutarylase, and a weak deacetylase [24–26]. SIRT5 can physically bind to some of its enzymatic targets, such as Mitochondrial Antiviral Signaling Protein MAVS [30], mitochondrial serine hydroxymethyltransferase SHMT2 [31], or pyruvate kinase PKM2 [32]. In these examples, SIRT5 both desuccinylates and binds the target protein as determined by co-immunoprecipitation. We thus hypothesized that SIRT5 could enzymatically modify Nsp14 and remove a putative succinyl, malonyl or glutaryl group.

To test this hypothesis, we determined if SIRT5 catalytic mutants bind Nsp14. Based on the structure of the SIRT5 catalytic domain and the homology with other sirtuins, we used or generated several expression constructs with mutations in conserved residues: H158 is catalytically required to abstract a proton from NAD, Q140 and I142 are involved in NAD binding, and Y102 and T105 interact with the extended acidic chains of succinyl or malonyl groups (Fig 2A). H158, Q140 and I142 are universally conserved in sirtuins, but Y102 and T105 are specific to SIRT5 and mediate the specificity to longer-chain acidic groups [23,33]. Mutation of these residues is known (for H158Y, Y102F and R105M) or predicted (for Q140A and I142A, based on homology with other sirtuins) to abolish SIRT5 desuccinylation activity. After transfection of Nsp14-strep and SIRT5 mutants in *SIRT5*-KD cells, the binding of SIRT5 to Nsp14 was lost or severely reduced in most mutants (Fig 2B). In particular, SIRT5 binding completely disappeared in H158Y and Q140A mutants, and only remained in significant amounts with the Y102F mutation. This result shows that an intact SIRT5 catalytic domain is necessary for the interaction with Nsp14. Of note, treatment of HEK293T cells with the proteasome inhibitor MG-132 did not affect the stability of the overexpressed catalytic mutants, which suggested that the different proteins could fold properly (S1A Fig).

To further establish that the catalytic activity of SIRT5 is necessary for the interaction with Nsp14, we used a recently described, potent and specific SIRT5 inhibitor (Sirt5-i) [34]. This inhibitor has a published $IC_{50}$ of 0.11 μM, and we measured an $IC_{50}$ of 0.44 μM in desuccinylation assays *in vitro* (S1B Fig). *SIRT5*-KD cells were transfected with Nsp14 and SIRT5 and incubated with increasing concentrations of Sirt5-i. The binding of SIRT5 was lost at high concentrations of Sirt5-i, with the interaction almost absent at 25 and 100 μM (Fig 2C). This observation again suggested that the SIRT5 catalytic activity was necessary for the interaction.

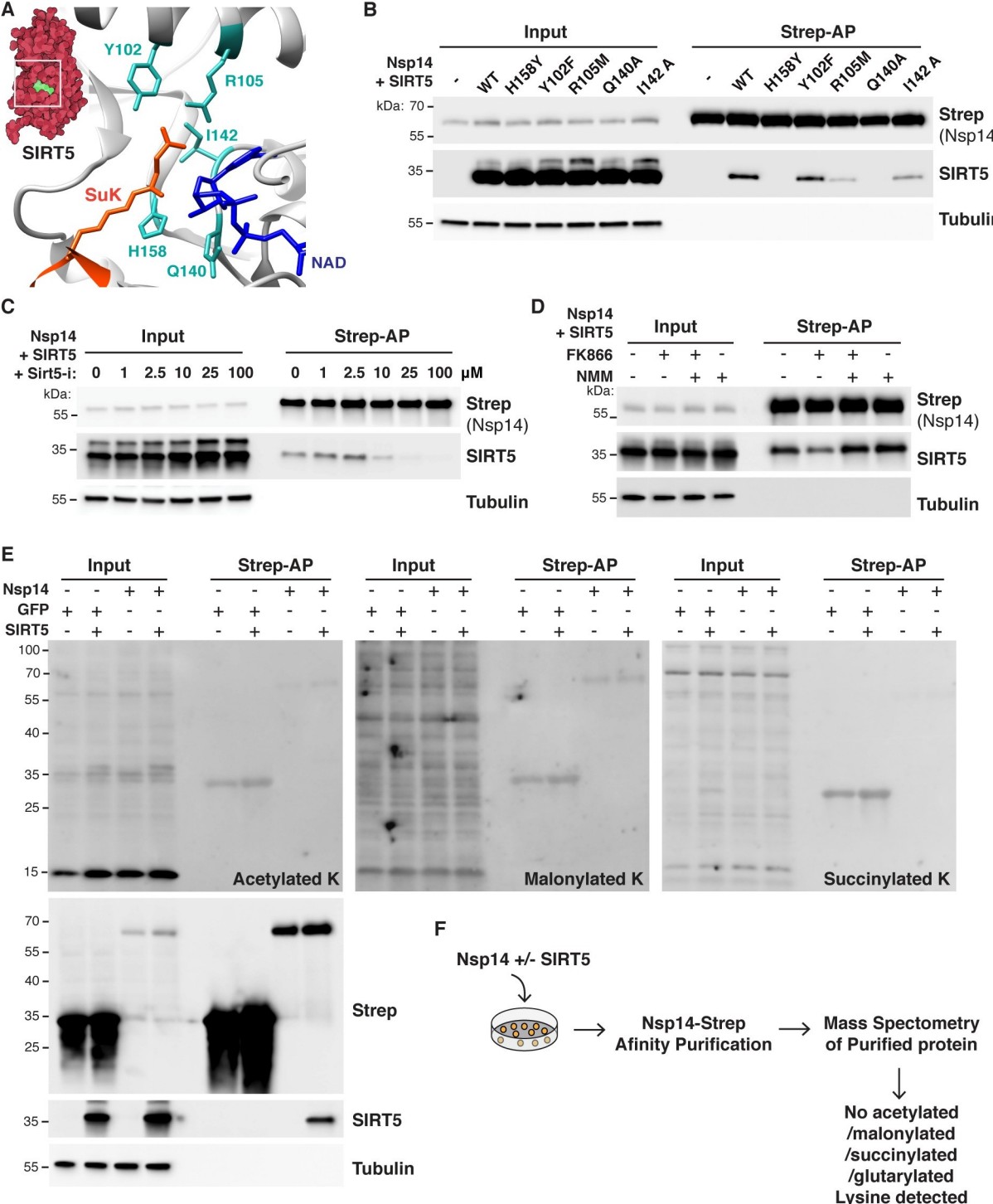

**Fig 2. SIRT5 catalytic activity is necessary to interact with Nsp14. A**. Cartoon representation of the protein structure of SIRT5 catalytic site in complex with cofactor NAD and succinylated lysine substrate (SuK), showing conserved residues mutated in panel B. **B**. Strep-tag affinity-purification and western blot after transfection of *Sirt5*-KD HEK293T cells with Nsp14-strep and SIRT5 catalytic mutants, showing that the interaction with Nsp14 is lost in several mutants. **C**. Strep-tag affinity-purification and western blot after transfection with Nsp14-strep and SIRT5, in *Sirt5*-KD HEK293T cells incubated with increasing concentrations of SIRT5 inhibitor Sirt5-i. High concentrations of Sirt5-i prevent the interaction. **D**. Strep-tag affinity-purification and western blot after transfection with Nsp14-strep and SIRT5, in *Sirt5*-KD HEK293T cells incubated with NAMPT FK866 inhibitor (low cellular NAD), FK866 and NMN, or NMN alone (high cellular NAD). SIRT5 binding strength correlated with NAD levels. **E**. Pan-acetylation, malonylation and succinylation in *SIRT5*-KD HEK293T total or Strep-purified proteins, after

transfection with Nsp14-Strep, GFP-strep control and/or SIRT5. No specific lysine modifications could be detected. **F.** Summary of mass spectrometry experiments. Nsp14-strep proteins purified from *SIRT5*-KD HEK293T, with or without co-transfection with SIRT5, were analyzed by mass spectrometry. No acetylation, malonylation, succinylation or glutarylation modifications could be detected.

Notably, until now, Sirt5-i had not been characterized in cell culture. This experiment suggested that Sirt5-i could be efficiently used in cells, with a putative $IC_{50}$ of approximately 10 μM. Next, we tested whether the interaction depended on cellular NAD levels. NAD is a necessary co-substrate, and SIRT5 activity is highly correlated with cellular NAD levels. In cells, most NAD is synthesized through the NAD salvage pathway [35]. Treating cells with the NAMPT inhibitor FK866 blocks the rate-limiting step in the pathway and rapidly depletes NAD levels, and supplementing cells with the NAD precursor nicotinamide mononucleotide (NMN) rescues the depletion [36] (S1C Fig). *SIRT5*-KD cells were transfected for 24 hours with Nsp14 and SIRT5 and for an additional 24 hours with FK866 and/or NMN (Fig 2D). SIRT5 binding was highly reduced in the presence of FK866 (low NAD levels), but the binding was rescued in presence of NMN, and appeared slightly stronger when cells were treated with NMN alone (high NAD levels). This finding indicated that the interaction of Nsp14 and SIRT5 positively correlated with cellular NAD levels. Altogether, by directly inhibiting SIRT5 or modulating the level of its co-substrate NAD, these experiments confirmed that SIRT5 catalytic activity was necessary for the interaction with Nsp14.

Finally, we determined if Nsp14 was directly modified by SIRT5 and if we could detect changes in the levels of acetylation, succinylation or malonylation. *SIRT5*-KD cells were transfected with SIRT5, Nsp14 and/or a GFP expression control. After affinity purification, pan-acetylation/succinylation/malonylation antibodies were used to detect changes in the levels of different lysine modifications (Fig 2E). Independent of the presence of SIRT5, we detected no measurable changes in acetylation, succinylation or malonylation, either among input or purified proteins. Weak bands were observed for purified GFP and Nsp14, but the intensity of the signal was not affected by the presence or absence of SIRT5, suggesting that it is not specific. Pan-modification antibodies are often not very sensitive, and we tested whether we could detect changes in lysine modifications by mass spectrometry. Nsp14-strep was transfected in *SIRT5*-KD cells, purified by affinity purification, and analyzed by mass spectrometry. The experiment was done with and without SIRT5 co-transfection, in three biological replicates each. In both conditions, Nsp14 had high coverage (65% coverage in presence of SIRT5, 88% without SIRT5), and we confirmed the interaction of Nsp14 with the two human proteins GLA and IMDH2 as reported [1] (S1 Data). However, despite 94% of lysine residues being covered (28 out of 30) and high quality of the data, we found no acetylated, succinylated, malonylated or glutarylated sites. We detected a previously characterized phosphorylation site at serine 56 [37] and a nitrosylation site on tyrosine 351. Since SIRT5 is the only known cellular desuccinylase, demalonylase, and deglutarylase, we had hoped that analyzing Nsp14 post-translational modifications in the presence or absence of SIRT5 would reveal changes in the level of these acylations marks. On the contrary, the absence of detectable acetylation, succinylation, malonylation and glutarylation suggested that Nsp14 was not directly modified by SIRT5. This series of experiments revealed that, even though SIRT5 catalytic activity is necessary for the interaction with Nsp14, Nsp14 did not appear to be a direct target of SIRT5.

## Nsp14 also interacts with SIRT1

The seven human sirtuins share conserved domains, and we tested whether Nsp14 also interacts with the other sirtuins. We found that Nsp14 interacted with SIRT1, but not with SIRT2, 3, 6 or 7 (Fig 3A). We could not investigate the interaction with SIRT4 because we lacked a

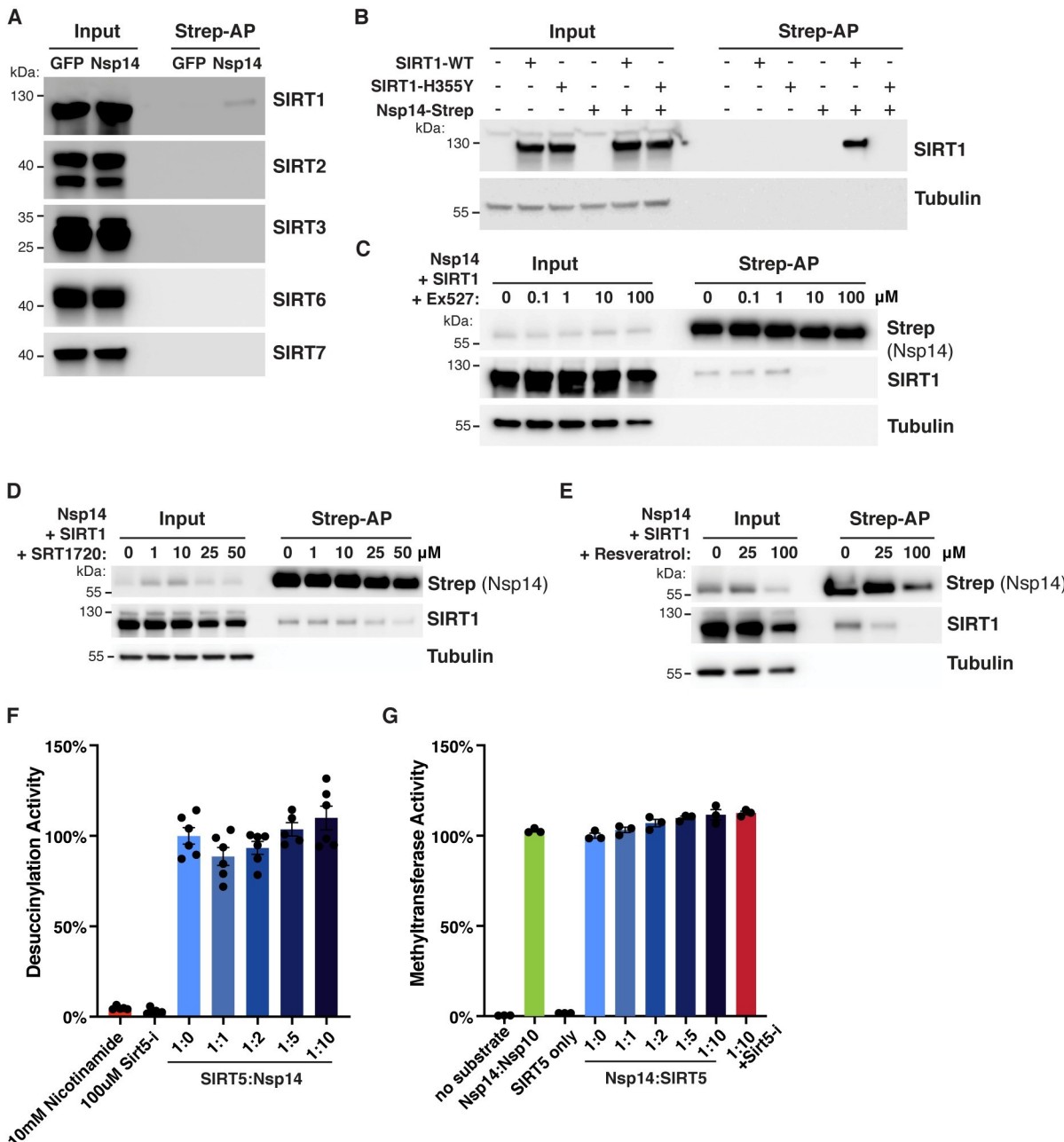

**Fig 3. SARS-CoV-2 Nsp14 interacts with human SIRT1. A**. Co-purification of endogenous sirtuins SIRT1, 2, 3, 6 and 7 after transfection of Nsp14-strep in HEK293T cells, as shown by western blot. Loading and purification controls are the same as in Fig 1B. **B**. Strep-tag affinity-purification and western blot after transfection of HEK293T cells with Nsp14-strep and SIRT1 WT and H355Y catalytic mutant, showing that the interaction with Nsp14 is lost in H355Y mutant. **C**. Strep-tag affinity-purification and western blot after transfection with Nsp14-strep and SIRT1, in HEK293T cells incubated with increasing concentrations of SIRT1 inhibitor Ex-527. High concentrations of Ex-527 prevent the interaction. **D-E**. Strep-tag affinity-purification and western blot after transfection with Nsp14-strep and SIRT1, in HEK293T cells incubated with increasing concentrations of SIRT1 specific activator SRT1720 (D) or non-specific activator resveratrol (E). Both drugs were cytotoxic at high concentrations and the apparent decrease in SIRT1 binding correlated with a similar decrease in the input lanes. **F**. *In vitro* desuccinylation activity of purified SIRT5 incubated with increasing concentrations of Nsp14, showing no effect. **G**. *In vitro* methyltransferase activity of purified Nsp14 incubated with increasing concentrations of SIRT5, showing no specific effect. Unmethylated GpppG cap-analog was used as a substrate.

specific antibody. The signal from SIRT1 was specific but appeared weaker than with SIRT5, which might explain why it was not detected by mass spectrometry [1]. Interestingly, and as we observed with SIRT5, mutating SIRT1 catalytic domain or treating cells with the SIRT1 inhibitor Ex-527 eliminated the interaction (Fig 3B and 3C). This finding suggested again that SIRT1 catalytic activity is necessary for the interaction with Nsp14. By contrast, treating cells with the specific SIRT1 activator SRT1720 or the non-specific activator Resveratrol had no measurable effect on Nsp14 binding (Fig 3D and 3E). Both activators were toxic to cells at high concentrations, and the apparent decrease in binding after affinity purification correlated with reduced levels of Nsp14 and SIRT1 in input lanes.

To uncover a putative molecular function of the Nsp14-SIRT5 interaction, Nsp14 and SIRT5 were expressed in *E. coli* and purified. Unfortunately, and even though the complex could be readily observed in mammalian cells, we could not reconstitute it *in vitro* with purified proteins, either by column chromatography or gel electrophoresis. This was the case when the two proteins were expressed separately and purified, or when co-expressed in *E. coli*. This observation suggests that the *in vitro* conditions that we used were inadequate and that additional parameters in mammalian cells were likely missing, such as other protein cofactors, post-translational modifications, or specific buffer conditions. Even though the complex could not be observed *in vitro*, we determined if the enzymatic activities of SIRT5 or Nsp14 were perturbed when in presence of each other. We measured the desuccinylation activity of SIRT5 alone or when incubated with increasing concentrations of Nsp14, using an *in vitro* desuccinylation assay. We detected no changes in desuccinylation activity in presence of Nsp14, whereas adding known inhibitors, such as nicotinamide or the specific SIRT5 inhibitor Sirt5-i, completely inhibited the activity (Fig 3F). Similarly, we used a methyltransferase assay to characterize the MTase activity of Nsp14 in presence of SIRT5. This assay measures the conversion of the methyl donor S-adenosyl methionine (SAM) into S-adenosyl homocysteine and can be used to measure the activity of any SAM-dependent methyltransferase. Nsp14 methylates the mRNA cap and Nsp14 was incubated with an unmethylated GpppG cap-analog, in presence of SAM, NAD and increasing concentrations of SIRT5 (Fig 3G). The Nsp14/Nsp10 complex and Nsp14 alone had similar activities, as predicted. We detected a small 10% increase of methyltransferase activity with excess SIRT5, but this increase persisted in presence of the SIRT5 inhibitor Sirt5-i. Since we showed above that Sirt5-i disrupted the interaction (Fig 3G), this small increase might not be specific. Besides, Nsp14 ExoN activity depends on Nsp10, and Nsp14 has no ExoN activity by itself. We showed that Nsp10 and SIRT5 are parts of separate complexes (Fig 1F and 1G), suggesting that the Nsp14/SIRT5 complex is very unlikely to have any ExoN activity either. Overall, these enzymatic assays failed to uncover a clear molecular function of the Nsp14/SIRT5 complex.

## SIRT5 is a proviral factor

We next wanted to examine the role of SIRT5 during SARS-CoV-2 infection. We generated a *SIRT5* knockout (*Sirt5*-KO) in A549 cells overexpressing ACE2 (A549-ACE2), using CRISPR-Cas9 editing and three commercially designed guide RNAs. The knockout was generated by transfecting cells with Cas9-gRNA ribonucleoproteins, and cells transfected with Cas9 alone and no guide RNA were used as wild-type (WT) controls. As expected, SIRT5 was undetectable in *SIRT5*-KO cells (Fig 4A). WT and *SIRT5*-KO cells were infected with SARS-CoV-2 (Wuhan strain, USA/WA-1/2020 isolate) at a multiplicity of infection (MOI) of 0.1 and 1, and viral RNA was quantified by RT-qPCR after 3 days (Fig 4B). At both MOIs, we observed a significant 2–3-fold decrease of SARS-CoV-2 mRNA in *SIRT5*-KO cells (p<0.0001 and p = 0.0002 at MOI = 0.1 and 1, respectively, by ANOVA). These results were confirmed by

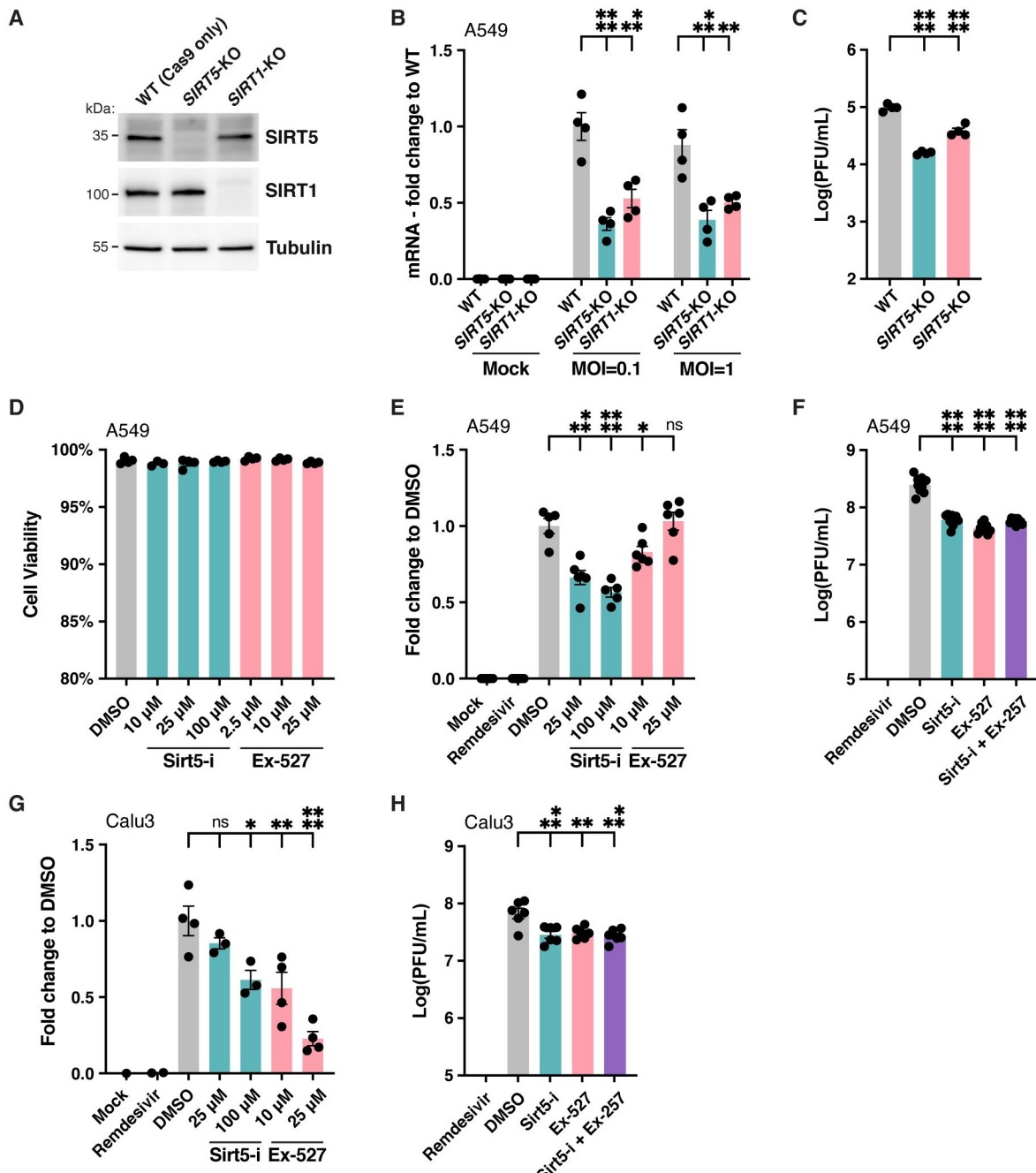

**Fig 4. SIRT5 is a proviral factor. A**. Western blot showing the absence of SIRT5 and SIRT1 in *SIRT5*- and *SIRT1*-KO A549-ACE2 cells, after CRISPR knockout. **B**. Decrease in cell-associated viral mRNA levels in *SIRT5*- and *SIRT1*-KO cells infected with SARS-Cov-2 for 3 days at MOI = 0.1 or MOI = 1, as shown by RT-qPCR. Data show fold-changes compared to WT levels at MOI = 0.1. n = 4. **C**. Decrease in viral titers in *SIRT5*- and *SIRT1*-KO cells infected with SARS-Cov-2 for 3 days at MOI = 1, as shown by plaque assay. n = 4. **D**. Absence of cytotoxicity in A549-ACE2 cells treated with Sirt5-i and Ex-527 inhibitor, as measured by flow cytometry. n = 4. **E**. Decrease in cell-associated viral mRNA levels in A549-ACE2 cells infected with SARS-Cov-2 for 3 days at MOI = 0.1, and treated with SIRT5 and SIRT1 inhibitors Sirt5-i and Ex-527, as shown by RT-qPCR. Data show fold-change compared to DMSO-treated levels. n = 6. **F**. Decrease in viral titers in A549-ACE2 cells infected with SARS-Cov-2 for 3 days at MOI = 0.1, and treated with SIRT5 and SIRT1 inhibitors Sirt5-i and Ex-527, as shown by plaque assay. n = 9. **G/H.** Same as E. (with n = 4), and F. (n = 6), using Calu3 cells. **B-H**. Data show mean and standard error of the mean (SEM) between biological replicates. RT-qPCR results were internally normalized with *GAPDH* and *ACTIN* reference genes. Viral titers after plaque assay are expressed in log-transformed PFU (plaque-forming unit) per mL of supernatant. Asterisks summarize the results of one-way ANOVAs followed by Holm–Šidák multiple comparisons test (on log-transformed data for plaque assays) *: p < 0.05, **: p < 0.01, ***: p < 0.001, ****: p < 0.0001.

plaque assay, and we measured a significant sixfold decrease of viral titers in *SIRT5*-KO cells (Fig 4C, p<0.0001, ANOVA). These observations suggested that SIRT5 is necessary for SARS-CoV-2 replication and/or propagation.

To confirm this result, A549-ACE2 cells were infected with SARS-CoV-2 in presence of the specific inhibitor Sirt5-i. Sirt5-i caused no measurable cytotoxicity (Fig 4D) and significantly reduced cell-associated viral mRNA levels by almost twofold (Fig 4E, p = 0.0001 and p<0.0001 at 25 and 100 μM, respectively, ANOVA). Viral titers measured by plaque assay were also significantly reduced by fourfold in presence of Sirt5-i inhibitor (Fig 4F, p<0.0001, ANOVA). Human lung-cancer cells Calu-3 endogenously express ACE2 and are often considered a better model for SARS-CoV-2 infection. Calu-3 cells were infected with SARS-CoV-2 and treated with SIRT5 inhibitor Sirt5-i. Viral RNA levels were significantly reduced, with a twofold reduction in viral mRNA at 100 μM (Fig 4G, p = 0.011, ANOVA). Viral titers measured by plaque assay were also significantly reduced by twofold in Calu3 cells in presence of Sirt5-i inhibitor (Fig 4H, p = 0.009). Altogether, this showed that knocking out or inhibiting SIRT5 resulted in a decrease in SARS-CoV-2 levels.

We obtained similar results with SIRT1. SARS-CoV-2 replication was reduced in *SIRT1*-KO A549-ACE2 cells, with a significant twofold decrease in mRNA levels (p = 0.0002 and p = 0.0011 at MOI = 0.1 and 1, respectively), and a significant twofold decrease in viral titers as well (Fig 4A, 4B and 4C, p<0.0001). SIRT1 inhibitor Ex-527 significantly reduced viral titers by sixfold in A549-ACE2 cells, and by twofold in Calu3 cells (p<0.0001 and p = 0.011, respectively, Fig 4E–4H). This showed that SIRT1 is also necessary for efficient SARS-CoV-2 infection. Interestingly, treating A549-ACE2 or Calu3 cells with both SIRT5 and SIRT1 inhibitors did not further reduce viral levels. Titers significantly decreased by two to fourfold in cells treated with Sirt5-I and Ex-527 together, a level similar to cells treated with the inhibitors alone (Fig 4F and 4H). This suggested that the roles of SIRT1 and SIRT5 during SARS-CoV-2 infection might be interdependent or that they act in the same pathway. In summary, our observations indicated that SIRT5 and SIRT1 are proviral factors necessary for SARS-CoV-2 replication and/or propagation. The reduction of viral levels without SIRT1 was less pronounced and consistent than without SIRT5, and we focused the rest of our investigation on SIRT5.

## SIRT5 proviral activity is partially independent of the interaction with Nsp14

To determine if the proviral role of SIRT5 could be explained by its interaction with Nsp14, we analyzed the role of SIRT5 during infection with human coronavirus HCoV-OC43, a distantly related beta-coronavirus. SIRT5 interacts with Nsp14 from SARS-CoV-2 and SARS-CoV, but not from MERS-CoV [2]. Similarly, we observed by affinity co-purification in HEK293T cells that SIRT5 was not interacting with Nsp14 from HCoV-OC43 (Fig 5A). This result further confirmed that SIRT5 interaction with Nsp14 is specific to SARS-like coronaviruses. Human colon adenocarcinoma cells HCT-8 were infected with HCoV-OC43 in presence of Sirt5-i inhibitor, and we observed a significant decrease in viral levels (Fig 5B and 5C). At 100 μM, viral mRNA in the cell-culture supernatant and viral titers measured by plaque assay were both reduced 10-fold (p = 0.0021 and p < 0.0001, respectively, ANOVA). This observation suggested that the role of SIRT5 during infection might be partially independent of its interaction with Nsp14.

## *SIRT5* knockout cells express a higher basal level of viral restriction factors

To gain insight into the role of SIRT5 during SARS-CoV-2 infection, we performed RNA-seq in A549-ACE2 cells, WT and *SIRT5*-KO, after 3 days of infection with SARS-CoV-2

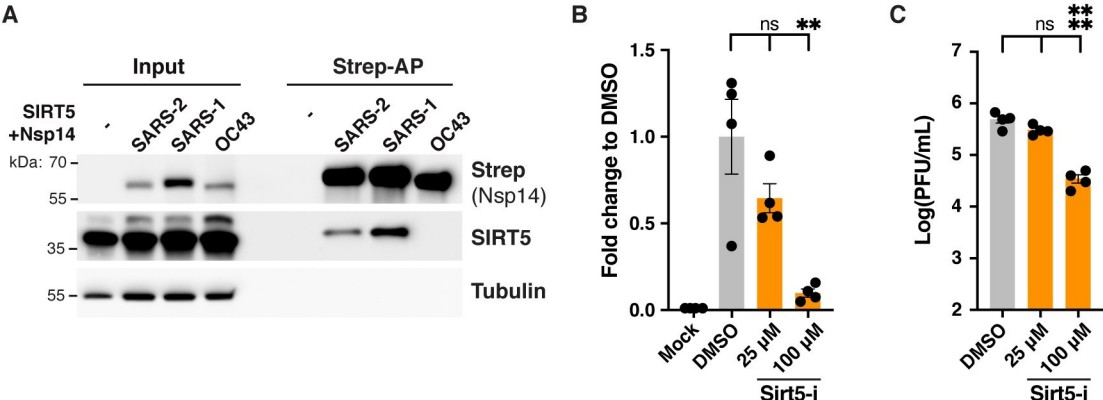

**Fig 5. SIRT5 proviral activity is partially independent from the interaction with Nsp14.** A. Strep-tag affinity-purification and western blot after transfection of *SIRT5*-KD HEK293T cells with SIRT5 and Nsp14-strep from different coronaviruses, showing that the interaction with SIRT5 is specific to SARS-like coronaviruses. B. Decrease in supernatant-associated viral mRNA levels in HCT-8 cells infected with HCoV-OC43 for 5 days at MOI = 0.1, and treated with SIRT5 inhibitor Sirt5-i, as shown by RT-qPCR. Data show fold-change compared to DMSO-treated levels. n = 4. C. Decrease in viral titers in HCT-8 cells infected with HCoV-OC43 for 5 days at MOI = 0.1, and treated with SIRT5 inhibitors Sirt5-i, as shown by plaque assay. n = 4. B-C. Data show mean and SEM between biological replicates. Asterisks summarize the results of one-way ANOVAs followed by Holm–Šidák multiple comparisons test (on log-transformed data for plaque assays) *: $p < 0.05$, **: $p < 0.01$, ***: $p < 0.001$, ****: $p < 0.0001$.

(MOI = 1). Sequencing was done in four biological replicates for each condition. Principal-component analysis showed that samples separated well, based on knockout and infection status (S2A Fig). When comparing uninfected WT and *SIRT5*-KO cells, 1139 and 869 genes were significantly up- and downregulated, respectively (q-value threshold q = 0.05). Most of these changes were modest, and only 69 genes were up- or downregulated by more than twofold (Fig 6A, left panel). Gene Ontology and pathway enrichment analysis showed that up-regulated genes were principally involved in metabolism and, in particular, protein catabolism processes in the lysosome, whereas down-regulated genes were involved in DNA replication and mitosis (S2 Data). These findings were in line with reports showing that SIRT5 is implicated in autophagy and controls cell proliferation in cancer cells by targeting multiple metabolic enzymes [31,32,38–41]. In WT infected samples, around 5% of total reads mapped to the SARS-CoV-2 genome, indicating substantial and successful viral replication. Despite this high level of viral expression, we observed a muted response to infection, at least in WT cells (Fig 6A, middle panel). SARS-CoV-2 efficiently evades innate immune defense through multiple mechanisms, and this absence of a strong transcriptional response is characteristic of SARS-CoV-2 infection and has been documented in numerous studies [42–44]. For example, we could not detect induction of *IFN-α*, *IFN-β*, *CXCL10*, interleukin *IL-6* or tumor necrosis factor (*TNF*). When comparing WT infected and WT mock-infected samples, 275 and 385 genes were significantly up- and down-regulated, respectively (threshold q = 0.05), with only the chemokine *CXCL8* (IL-8) and the transcription factor *ATF3* being increased with a fold-change higher than 2 (Fig 6A, middle). Gene Ontology and pathway enrichment analysis indicated that genes involved in response to virus infections were upregulated, with the two most upregulated pathways being the NOD-like and RIG-I-like receptors signaling pathways, which are implicated in the intracellular recognition of viruses (S2 Data). Notably, other studies that reported a higher number of differentially expressed genes often had higher levels of viral infection, with a fraction of SARS-CoV-2 reads of 10–50%, compared to 5% in this study. This might explain why the transcriptional response that we observed is comparatively smaller [42,45].

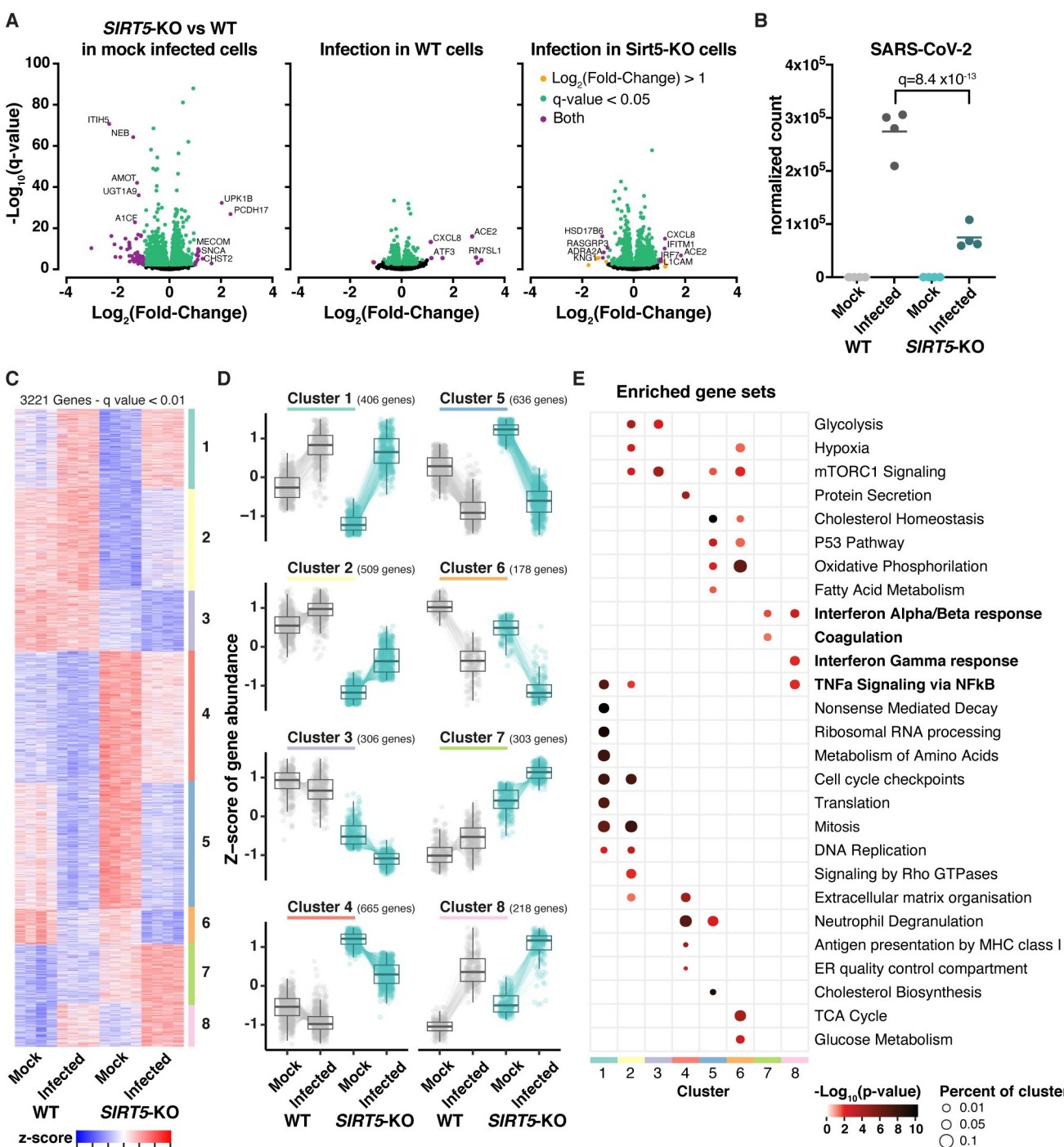

**Fig 6. *SIRT5*-KO cells mount a stronger innate immune response.** RNA-seq analysis of WT and *SIRT5*-KO A549-ACE2 cells infected or mock-infected for 3 days with SARS-CoV-2 at MOI = 1. n = 4. **A**. Volcano plots showing differentially expressed genes between the different conditions. Highlighted genes display a q-value q<0.05 (green), log2 fold-change >1 (orange), or both (purple). Left panel: *SIRT5*-KO vs WT in mock-infected cells. Middle: Infected vs mock-infected WT cells. Right: Infected vs mock-infected *SIRT5*-KO cells. **B.** Normalized gene count of SARS-CoV-2. **C-D.** Unsupervised clustering of the 3221 genes differentially expressed between at least two of the four conditions (q<0.01). **C:** heatmap of normalized expression. **D.** Z-scores of differentially expressed genes as grouped by clustering. Colored lines represent the quantification of an individual gene whereas solid black lines show the cluster Tukey boxplot. **E.** Enrichment analysis of biological gene sets in the identified gene clusters (C and D).

We next analyzed the effect of *SIRT5* knockout on SARS-CoV-2 infection. SARS-CoV-2 levels were almost fourfold less in *SIRT5*-KO infected cells than in WT infected cells, confirming that SIRT5 is a proviral factor (q = 8.4 x10$^{-13}$, Fig 6B). We focused our analysis on the 3221 genes that were differentially expressed between at least two of the four conditions (threshold q = 0.01). Hierarchical consensus clustering of the differentially expressed genes generated eight clusters, representing groups of genes that behaved similarly between the different sample conditions (Fig 6C and 6D). Enrichment analysis of biological gene sets then allowed the identification of the cellular pathways over-represented in the identified clusters (Fig 6E). For example, clusters 2 and 3 corresponded to genes with a lower expression in *SIRT5*-KO cells, independently of the infection status, and pathways linked to the cell cycle and cellular metabolism were significantly enriched in these clusters. In most of the clusters (clusters 2–6), pathways linked to cellular metabolism were significantly enriched, highlighting that SIRT5 is an important metabolic enzyme. Clusters 7 and 8 were particularly interesting. They represent genes that are expressed at a higher basal level in *SIRT5*-KO cells, and whose expression is further increased during infection (Fig 6C and 6D). Strikingly, pathways linked to innate immunity, such as type I and II interferon and NFκB signaling, were significantly enriched in these clusters (Fig 6E). Genes in clusters 7 and 8 are up-regulated in *SIRT5*-KO cells, even in uninfected cells, which suggested that *SIRT5*-KO cells had a higher basal level of innate immunity markers and could mount a stronger immune response.

We thus investigated whether innate immunity pathways were up-regulated in *SIRT5*-KO cells, even without viral infection. In mock-infected cells, the Gene Ontology term "Innate Immune Response" was significantly enriched in *SIRT5*-KO samples (q = 0.0071, S2 Data). Type-I interferon responses are one of the most important lines of defense against viruses. We analyzed the expression of known Type-I interferon-stimulated genes, as well as of other genes broadly linked to innate immune responses and present in clusters 7 and 8 (Fig 7A). As could be expected, most of these genes were upregulated in infected cells. Strikingly, many of these genes, which are normally involved in the response to pathogens, were also upregulated in *SIRT5*-KO cells in absence of infection, with 39 out of the 71 selected genes being significantly upregulated in mock-infected *SIRT5*-KO cells (q < 0.05). Many of these genes were expressed at similar levels between infected WT cells and mock-infected *SIRT5*-KO cells, and were further increased in infected *SIRT5*-KO cells (Fig 7A). This included cytokines (*IL33*, *CXCL5*, and *CSF1*), the transmembrane restriction factors *IFITM2* and 3, members of the complement system (*C1S*, *C1R* and *C3*), the MHC class I subunit *B2M*, and other interferon-stimulated genes or restriction factors (e.g., *IFIT3*, *TRIM22*, *STAT3*, *MMP7*, *LCN2*, *MUC5AC*, *SLFN5*, *NT5E*, *CAST* and *SNCA*) [46–52]. The upregulation between WT and *SIRT5*-KO in mock-infected cells was modest but statistically significant, ranging from 20 to 50%, and was further increased up to twofold in infected *SIRT5*-KO cells (Figs 7A and S2B). These results suggest that *SIRT5*-KO cells express a higher basal level of numerous restriction factors and mount a stronger antiviral response, which could explain why SARS-CoV-2 levels are decreased in absence of SIRT5.

To confirm these observations, we validated by RT-qPCR the upregulation of several restriction factors between WT and *SIRT5*-KO cells, in absence of infection (Fig 7B). This confirmatory analysis was done with new samples and independently of the RNA-seq experiment, using eight biological replicates per group to increase statistical power and measure subtle effects. The cytokines *IFN-β* and *IL33* were significantly upregulated two to threefold in *SIRT5*-KO cells (p = 0.0004, and p<0.0001, respectively, t-test). Other markers such as *STAT3*, *IFIT3* and *B2M* showed modest but statistically significant upregulation ranging from 20 to 70%, while *IFITM2* and *SCNA* had increased expression that did not reach statistical significance. These results confirmed that *SIRT5*-KO cells express a higher basal level of viral restriction factors.

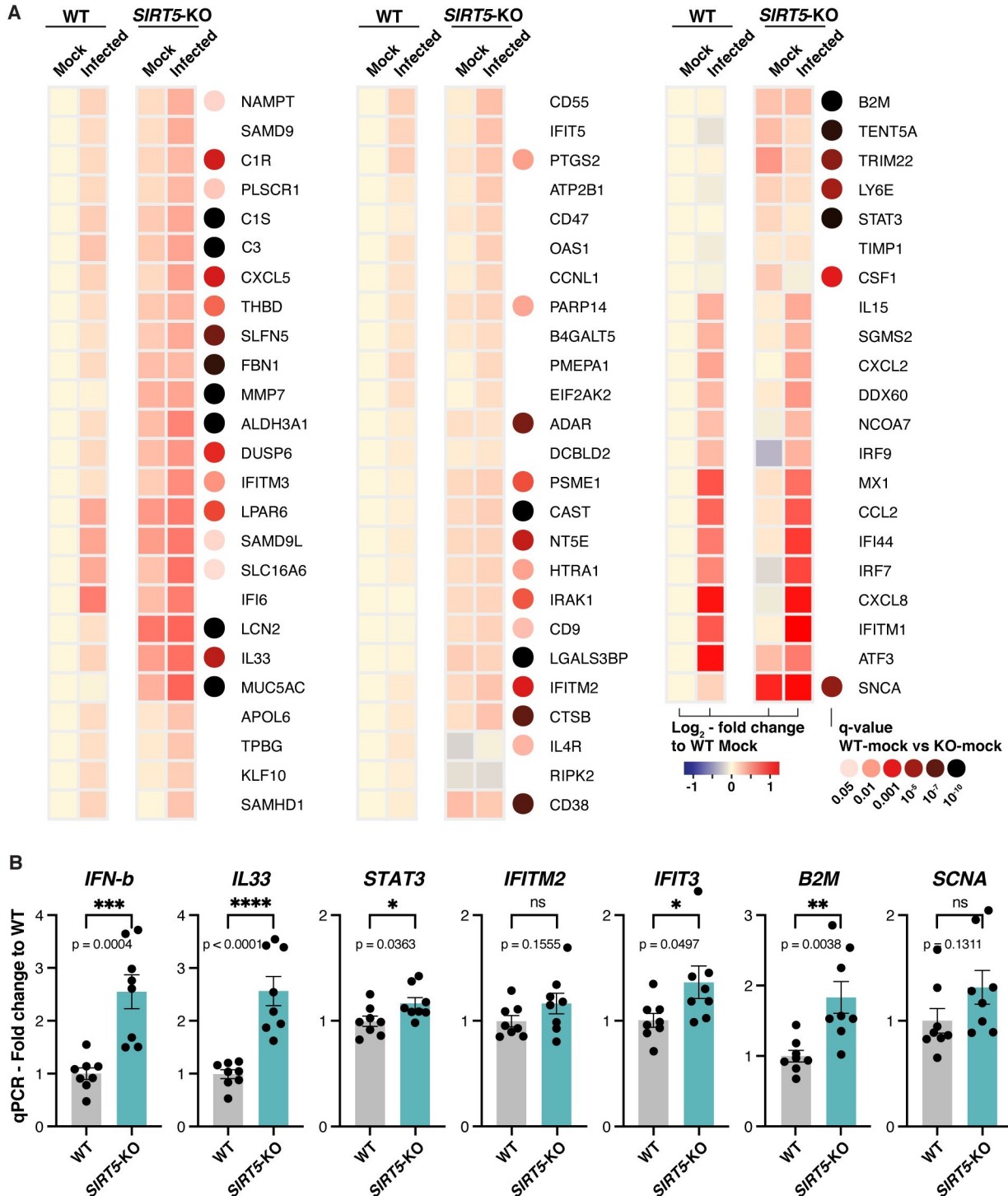

**Fig 7. *SIRT5*-KO cells express a higher basal level of viral restriction factors. A**. Expression heatmap of interferon-stimulated genes and other restriction factors, showing that mock-infected *SIRT5*-KO cells express higher basal levels of restriction factors, and that antiviral responses are stronger in *SIRT5*-KO cells. Data show mean log2 fold-change, compared to mock-infected WT, and the q-value between mock-infected WT and *SIRT5*-KO cells. Only genes differentially expressed between at least two conditions were included in the analysis (q<0.01). **B**. RT-qPCR confirmation of restriction factors upregulated in non-infected *SIRT5*-KO cells (n = 8). Data show fold-changes compared to WT levels after normalization with *ACTIN*. Data show mean and SEM between replicates. p-values after unpaired two-tailed t-test are shown and asterisks summarize the results. *: p < 0.05, **: p < 0.01, ***: p < 0.001, ****: p < 0.0001.

## SIRT5 proviral activity is independent of the MAVS signaling pathway

SIRT5 is implicated in RIG-1/MAVS antiviral signaling, a key innate immune pathway that recognizes viral RNA in the cytosol and activates type I interferon. Specifically, SIRT5 desuccinylate MAVS and reduce MAVS aggregation on the mitochondrial surface, resulting in lower interferon activation (S3A Fig) [30]. To explain the reduced viral replication in *SIRT5*-KO cells, we hypothesized that SIRT5 absence could lead to stronger MAVS aggregation and in turn a stronger antiviral response. To test this hypothesis, we used CRISPR-Cas9 to delete MAVS in A549 cells stably co-expressing ACE2 and TMPRSS2 (A549-A/T cells). A549-A/T cells can be infected more efficiently and represent an improvement over cells expressing ACE2 only, and were used for this experiment. Cells were transduced with a lentiviral vector expressing Cas9 and a gRNA against *MAVS*, which resulted in a 90% knockdown of MAVS by western blot (Fig 8A). *MAVS* knockdown (*MAVS*-KD) cells were then infected with

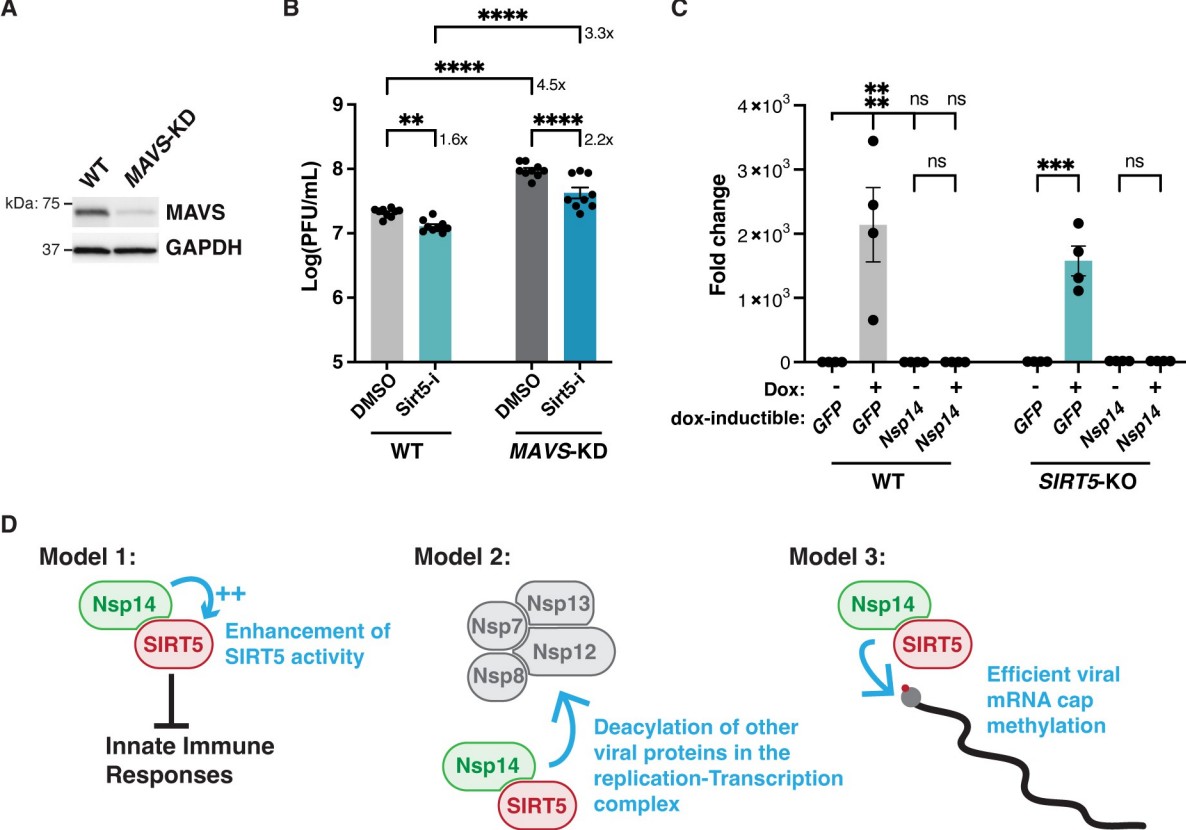

**Fig 8. SIRT5 proviral activity is independent of the MAVS signaling pathway. A**. Western blot showing 90% reduction of MAVS in A549-A/T cells transduced with a CRISPR lentivirus against MAVS. A549-A/T cells stably co-express ACE2 and TMPRSS2. **B**. Viral titers in *MAVS*-KD cells treated with DMSO or SIRT5 inhibitor Sirt5-i, after infection with SARS-Cov-2 for 3 days at MOI = 0.1, as shown by plaque assay. Sirt5-i had a similar effect in WT and *MAVS*-KO, suggesting that SIRT5 function is independent of MAVS. n = 9. **C**. RT-qPCR of *GFP* or *Nsp14* after doxycycline induction. WT and *SIRT5*-KO A549-ACE2 cells were stably transduced with doxycycline-inducible constructs for GFP and Nsp14. After doxycycline treatment for 48 hours at 100ng/mL, *GFP* was strongly overexpressed, but *Nsp14* failed to be expressed. Data show fold-changes compared to the first column (WT cells transduced with GFP without doxycycline), after normalization with *ACTIN*. n = 4. **D.** Hypotheses for the role of the SIRT5/Nsp14 interaction during SARS-CoV-2 infection. In model 1, Nsp14 could enhance SIRT5 activity, which would decrease innate immune responses and favor viral replication. In model 2, Nsp14 could redirect SIRT5 to novel targets, potentially in the replication-transcription complex, where SIRT5 could deacylate other viral proteins. In model 3, the Nsp14/SIRT5 complex could be primarily involved in mRNA cap methylation. Absence or inhibition of SIRT5 would lead to incomplete cap methylation and stronger immune recognition of viral mRNA. **B-C**. Data show mean and SEM between biological replicates. Asterisks summarize the results of one-way ANOVAs followed by Holm–Šidák multiple comparisons test (on log-transformed data for plaque assays). *: $p < 0.05$, **: $p < 0.01$, ***: $p < 0.001$, ****: $p < 0.0001$.

SARS-CoV-2 in presence or absence of SIRT5 inhibitor (Fig 8B). *MAVS* knockdown resulted in a three to five-fold increase in viral levels in both DMSO or Sirt5-i-treated cells (p<0.0001, ANOVA), indicating that MAVS is implicated in the antiviral response against SARS-CoV-2, independently of SIRT5. If SIRT5 was acting primarily by reducing MAVS activation, inhibiting SIRT5 in *MAVS*-KD cells would be expected to have little effect. However, we found that inhibiting SIRT5 had the same effect in WT and *MAVS*-KD cells, with a significant reduction of titers of about twofold in both WT and *MAVS*-KD cells (p = 0.0052 and p<0.0001, respectively, ANOVA). This important result invalidated our hypothesis and suggested that SIRT5 role in innate immunity is not limited to regulating MAVS. In fact, this showed that SIRT5 function during SARS-CoV-2 infection is likely independent of the RIG-I/MAVS pathway.

Finally, we wanted to elucidate whether Nsp14 played a role in modulating SIRT5 proviral activity. A549 cells cannot be transfected efficiently, and we attempted to generate cell lines expressing Nsp14. Nsp14 is cytotoxic, and we could not build cell lines stably expressing Nsp14 from a constitutive lentiviral vector. To circumvent this toxicity issue, we used doxycycline-inducible Nsp14 and GFP lentiviral vectors. A549-ACE2 WT and *SIRT5*-KO cells were transduced with inducible Nsp14 and GFP constructs, and cells were successfully selected with puromycin. In absence of doxycycline, cells stably transduced with inducible Nsp14 had no apparent replicative or morphological defects. As expected, doxycycline treatment induced a strong overexpression of *GFP*, with a 2000-fold induction compared to basal levels as measured by RT-qPCR (Fig 8C). However, we did not observe any change in *Nsp14* expression after doxycycline treatment. The continuous puromycin selection ensured that the lentivirus vector had been stably integrated, but *Nsp14* induction was nonetheless not functional. It is possible that leaky basal expression of *Nsp14* in absence of doxycycline was sufficient to negatively select against permissive integration events. Unfortunately, these results prevented us from further investigating if Nsp14 could impact SIRT5 activity in cells.

## Discussion

In this study, we investigated the role of SIRT5 during SARS-CoV-2 infection. Our results show that SIRT5 interacts with the non-structural viral protein Nsp14, and that this interaction is independent of Nsp10. Interestingly, we found that the catalytic activity of SIRT5 was necessary for the interaction, as several SIRT5 catalytic mutants could not bind to Nsp14, and the interaction was blocked by a specific SIRT5 inhibitor. SIRT5 is the main cellular desuccinylase, demalonylase, and deglutarylase, but we could not detect these lysine modifications on Nsp14 protein, suggesting that Nsp14 is not directly modified by SIRT5. We further showed that SIRT5 is a proviral factor and that SARS-CoV-2 levels decrease when SIRT5 is deleted or inhibited in cell-culture experiments, independently of the MAVS signaling pathway. We observed that *SIRT5*-KO cells express innate immunity markers at a higher basal level and mount a stronger antiviral response, which might explain the decrease in viral levels. Taken together, our study uncovered a novel and unexpected role for SIRT5 during SARS-CoV-2 infection.

The interaction between SIRT5 and Nsp14 is intriguing. Mutating the SIRT5 catalytic domain or treating cells with a SIRT5 inhibitor blocked the interaction (Fig 2B and 2C), and the strength of the interaction appeared to be modulated by cellular NAD levels (Fig 2D). NAD is necessary for SIRT5 activity, and these results highlight that SIRT5 catalytic activity or at the very least a functional catalytic domain is necessary to interact with Nsp14. SIRT5 binds to some of its targets in co-purification experiments, such as with MAVS, SHMT2, or PKM2 [30–32]. However, in these examples, SIRT5 was also desuccinylating the target proteins. Here we found no lysine modifications on Nsp14, either by mass spectrometry or immunoblotting.

Our experiments might not have been sensitive enough to detect it, but our mass spectrometry data had very high purity, covered 94% of Nsp14 lysine residues, and our pipeline routinely detects such modifications. Besides, SIRT5 is the only known desuccinylase, demalonylase, and deglutarylase, and these experiments were performed in *SIRT5*-KD cells, which would have enriched the presence of these lysine modifications if they were present. The unusual nature of the interaction might explain why we were unable to reconstitute the complex *in vitro* with proteins purified in *E. coli*, and why our enzymatic assays failed to uncover a clear molecular function. Interestingly, an interaction of a similar nature has been described between SIRT1 and the HIV viral protein Tat [53,54]. Tat interacted with the SIRT1 catalytic domain and mutation of conserved residues disrupted the interaction. In this case however, Tat was deacetylated by SIRT1 and blocked SIRT1 activity. Here, we could not determine if SIRT5 activity was altered by Nsp14, and global succinylation and malonylation levels appeared unaffected as well (Fig 2E). Further studies will be necessary to understand the nature of the interaction and its function. They could reveal a novel interaction mechanism, in which an enzyme binds to a partner through its catalytic domain, which has rarely been observed.

The role of SIRT5 and other sirtuins in disease is unclear. SIRT5, by modulating key metabolic enzymes, could be involved in cancer [31,32,38–41]. *SIRT5*-KO mice display no obvious phenotype and mount strong innate immune responses against several bacterial infections [55]. The role of SIRT5 and other sirtuins during viral infection is understudied and likely depends on the pathogen. Knocking out SIRT5 enhances the replication of several DNA viruses, such as herpes simplex 1, human cytomegalovirus and adenovirus type 5, and the same study reported a potential increase in influenza replication, albeit non-significantly [56]. By contrast, here we showed that inhibition or deletion of SIRT5 led to a decrease in viral levels with two distinct coronaviruses, SARS-CoV-2 and HCoV-OC43, indicating that SIRT5 is a proviral factor in this context (Figs 4 and 5). A recent study showed that replication of vesicular stomatitis virus (VSV) and Sendai virus (SeV), two negative-strand RNA viruses, was also diminished in absence of SIRT5 [30]. Whether SIRT5 broadly acts as a restriction factor against DNA viruses and as a proviral factor against RNA viruses will be an interesting hypothesis to investigate in the future.

The decrease of SARS-CoV-2 levels in *SIRT5*-KO cells correlated with elevated basal levels of numerous viral restriction factors, even in mock-infected cells (Fig 7). This upregulation was modest but highly significant and may account for why SARS-CoV-2 propagated slower in *SIRT5*-KO cells. Several hypotheses could explain this elevation of innate defenses. SIRT5 has been directly implicated in the RIG-I/MAVS pathway, a critical component of the innate sensing of RNA virus (S3A Fig) [30]. Recognition of cytosolic RNA molecules by RIG-I-like receptors, including RIG-I and MDA5, causes recruitment of MAVS, which forms large aggregates on the surface of mitochondria, ultimately leading to type I IFN production [57]. MAVS is succinylated upon viral challenge with VSV and SeV, and desuccinylation of MAVS by SIRT5 diminishes MAVS aggregation, resulting in lower interferon activation. By preventing MAVS aggregation, SIRT5 therefore enhances viral replication, at least with VSV and SeV [30]. However, by infecting *MAVS*-KD cells in presence of SIRT5 inhibitor, here we showed that SIRT5 likely acts independently of MAVS during SARS-CoV-2 infection (Fig 8B). Besides, SARS-CoV-2 efficiently counteracts MAVS activation of innate defense. The coronavirus proteins M, Orf9b and Orf3b directly antagonize MAVS aggregation or downstream signaling [58–60], and MAVS knockout was reported to have no effect on SARS-CoV-2 replication [43]. Our results indicated otherwise, but altogether this suggests that the desuccinylation of MAVS by SIRT5 is not the main mechanism for explaining the decrease of viral levels in *SIRT5*-KO cells. Other mechanisms could be involved. For example, SIRT5 is involved in the detoxification of reactive oxygen species (ROS) and attenuates cellular ROS levels [27]. Elevated ROS

levels activate innate immune responses, and the absence of SIRT5 could cause the activation of innate immune responses through this pathway [61]. Furthermore, SIRT5 regulates proteins involved in glycolysis, the TCA cycle, and fatty acid oxidation. These pathways were impaired without SIRT5, as well as mTOR signaling (Fig 6E). mTOR and the cellular sensing of energy and nutrient levels can directly influence immune activity [62], highlighting another pathway that could lead to immune activation in absence of SIRT5.

The potential link between our two key findings, namely the Nsp14/SIRT5 interaction and the proviral role of SIRT5, will be the subject of further investigation. Several hypotheses can be considered (Fig 8D). First, Nsp14 might work by enhancing SIRT5 activity, which would favor viral replication by dampening innate immune responses. In this model, increased deacylation of cellular targets of SIRT5 could result in a weaker immune response and favor viral replication. Second, Nsp14 could redirect SIRT5 toward novel targets, for example other viral proteins. Nsp14 is part of the viral replication-transcription complex, and SIRT5 could be involved in the deacylation of other members of the complex such as Nsp7, Nsp8, Nsp12 or Nsp13. Third, we observed that SIRT5 and Nsp10 were part of separate complexes, and that Nsp14 MTase activity increased slightly in presence of SIRT5 (Fig 3G). SIRT5 and Nsp10 could be separately involved in the two enzymatic activities of Nsp14, with the Nsp14/SIRT5 complex primarily responsible for viral mRNA Cap methylation. Absence of SIRT5 would result in a defect in cap methylation, more efficient immune recognition of viral RNA, and stronger immune response, as we observed.

To summarize, further studies will be necessary to elucidate how SIRT5 enhances SARS-CoV-2 replication, and how the interaction with Nsp14 plays in this context. Potent inhibitors of SIRT5 are in development, and SIRT5 is a potential target against cancer [34,40,63]. Our manuscript highlights that SIRT5 is a potential pharmaceutical target that could help against viral infections as well, and SARS-CoV-2 in particular. Currently, very few antiviral drugs are approved. Inhibiting SIRT5 will probably never represent a first line of defense, but it could be used in combination with drugs that directly target viral enzymes, leading to novel therapeutic regimens against COVID-19.

## Materials and methods

### Mammalian cell lines and culture conditions

All cell lines were maintained at 37°C in a 5% $CO_2$ humidified incubator. Cells were frequently tested for mycoplasma contamination and consistently tested negative.

HEK293T cells were obtained from ATCC (Cat. CRL-11268) and maintained in high glucose Dulbecco's minimal Eagle's medium (DMEM) with 10% fetal bovine serum (FBS) (Sigma-Aldrich, USA) and 100 μm/L penicillin-streptomycin (Corning, USA). Calu3 cells were obtained from ATCC (Cat. HTB-55) and cultured in AdvancedMEM (Gibco, USA), supplemented with 2.5% FBS (GeminiBio, USA), 1% GlutaMax, and 100 μm/L penicillin-streptomycin. Wild-type A549 cells were purchased from ATCC (Cat. CCL-185). A549 cells stably expressing ACE2 (A549-ACE2) were a gift from O. Schwartz (Pasteur Institute, Paris). A549-ACE2 cells were cultured in DMEM, supplemented with 10% FBS (GeminiBio, USA) and blasticidin (20 μg/ml) (Sigma-Aldrich, USA). Short-terminal repeat (STR) analysis by the Berkeley Cell Culture Facility on 17 July 2020 authenticated these as A549 cells with 100% probability. Vero-E6 cells were obtained from ATCC (Cat. CRL-1586) and cultured in DMEM (Corning, USA), supplemented with 10% FBS (GeminiBio, USA), 1% glutamine (Corning), and 100 μm/L penicillin-streptomycin (Corning). HCT-8 cells (ATCC Cat. CCL-244) were cultured in DMEM with 10% FBS and penicillin-streptomycin.

Small-molecule inhibitors used in the study are listed in S3 Data.

## Plasmids

Plasmids expressing GFP and Nsp14 proteins (from SARS-CoV-2, SARS-CoV and HCo-V-OC43) with a C-terminus strep tag were a gift from Nevan Krogan [1,2], and are also available on Addgene (pLVX-EF1alpha-SARS-CoV-2-nsp14-2xStrep-IRES-Puro, Addgene #141380). Nsp10-Flag plasmid was a gift from the Ott lab. Doxycycline-inducible plasmids were also a gift from the Krogan lab.

Mammalian expression plasmids for SIRT5 and SIRT5-H158Y with a myc-his tag in a pCDNA 3.1 vector were available in the Verdin lab [26]. Y102F, R105M, Q140A and I142A mutants were generated by Q5 site-directed mutagenesis (NEB, USA).

## Generation of CRISPR cell lines

Stable cell lines transduced with lentiviral vectors were generated as follow: lentiviruses expressing the constructs of interest were produced in HEK293T cells by standard protocols [64]. Cells were plated and transduced 24h later at low MOI when 50–80% confluent. Puromycin, hygromycin or other selection antibiotics were added 48h later and cells passaged at least three time in presence of antibiotics before being used for experiments.

The *SIRT5* knockdown (*SIRT5*-KD) cell line was generated using CRISPR interference in HEK293T cells. First, we generated a stable cell line expressing dCas9-KRAB-MeCP2. Early passage HEK293T cells were transfected with 1.5 μg of dCas9-KRAB-MeCP2 repressor plasmid (Addgene #110824) and 0.5 μg of Piggyback Transposase (gift from Maxim Greenberg), using PEI 25K transfection reagent (Polysciences Inc, Cat. 23966–1), according to the manufacturer's instructions. Cells stably expressing dCas9-KRAB-MeCP2 were selected with Blasticidin (Invivogen, USA) for 10 days before generating *SIRT5*-KD cells. Second, sequences for sgRNA against human *SIRT5* (5'- GGCGCTCCGGACCTGAGCCA-3') or non-targeting sgRNA (5'-GCTGCATGGGGCGCGAATCA-3') were obtained from Horlbeck et al. [65] and cloned into Addgene #84832 by annealing and ligation using T4 ligase. Plasmids were validated by Sanger sequencing (Elim Biopharmaceuticals. HEK293T cells expressing dCas9-KRAB-MeCP2 were transduced with lentiviruses expressing the gRNAs in medium containing 1 μg/ml polybrene and 30% lentivirus-containing supernatant (v/v). Cells were then selected with Blasticidin at 5 μg/mL and Puromycin (1 μg/mL, Invivogen, USA) to select for cells stably expressing both dCas9 and the sgRNAs. Stable knockdowns were validated by western blot.

The *SIRT5* knockout (*SIRT5*-KO) cell line was generated using CRISPR-Cas9 editing in A549-ACE2 cells. sgRNAs were commercially designed by Synthego (Redwood, California, USA) and are designed to work cooperatively to generate small-fragment deletions in early exons causing knockout (S3 Data). We combined 10 pmol of *Streptococcus pyogenes* NLS-Sp. Cas9-NLS (SpCas9) nuclease (Aldevron, USA, Cat. 9212) with 30 pmol of total synthetic sgRNA (10 pmol each sgRNA) to form ribonucleoproteins (RNPs) in 20 μL of total volume with SE Buffer for A549-ACE2 cells. The RNP assembly reaction was mixed by pipetting up and down and incubated at room temperature for 10 minutes. Cells were resuspended in transfection buffer, according to cell type, added to the preformed RNP solution, and gently mixed. Nucleofections were performed on a Lonza nucleofector system (Lonza, Switzerland, Cat. AAU-1001), using program CM-120 for A549-ACE2 cells. Cells were grown for several passages and genotyped by PCR and Sanger sequencing to confirm efficient editing. Absence of SIRT5 was confirmed by western blot.

A549 cells stably co-expressing ACE2 and TMPRSS2 (A549-A/T) were generated through sequential transduction of A549 cells with TMPRSS2-encoding (generated using Addgene plasmid #170390, a gift from Nir Hacohen) and ACE2-encoding (generated using Addgene

plasmid #154981, a gift from Sonja Best) lentiviruses and selection with hygromycin (250 μg/mL) and blasticidin (20 μg/mL) for 10 days, respectively. ACE2 and TMPRSS2 expression was verified by western blot. CRISPR/Cas9-mediated deletion of MAVS was accomplished using lentiviral transduction. A gRNA specific to the third exon of the *MAVS* gene was designed using Benchling.com and cloned into the lentiCRISPR v2 plasmid (Addgene #52961, gRNA sequence: 5'-GCTGGTAGCTCTGGTAGACAG-3'). A549 A/T were transduced with lentiviruses packaged with control (annotated WT) or MAVS-targeting gRNAs in the presence of polybrene (Sigma, TR-1003-G). Cells were selected with Puromycin for seven days and MAVS reduction was validated by western blot.

## Transfection, strep affinity purification, and Flag immunoprecipitation in HEK-293T cells

HEK-293T cells were plated in six-well plates or 10-cm dishes. After 24 hours, cells were transfected using polyethylenimine (PEI). Nsp14 is cytotoxic, and we used 0.5 μg of Nsp14-strep plasmid for a six-well plate and 4 μg for a 10-cm dish. Other co-transfecting plasmids, such as pcDNA-SIRT5, were used at the same concentration except when specifically mentioned. The total amount of plasmid was normalized using empty vectors when necessary. Plasmids were complexed with PEI in Opti-MEM medium (Thermofisher) at a 1:3 ratio, and the mixture was deposited onto cells dropwise. After 48 hours, cells were washed once with PBS, scraped off the plate by thorough pipetting, pelleted by centrifugation at 200g and 4˚C for 3 minutes, and frozen at −80˚C.

Affinity purification followed the methods of Gordon *et al.* 2020 [1]. Frozen cell pellets were thawed on ice and resuspended in 0.5 ml of lysis buffer (IP buffer: 50 mM Tris-HCl, pH 7.4, 150 mM NaCl, and 1 mM EDTA), supplemented with 0.5% Nonidet P40 substitute (NP40; Fluka Analytical) and cOmplete mini EDTA-free protease and PhosSTOP phosphatase inhibitor cocktails (Roche)). Samples were frozen on dry ice for 10–20 minutes and partially thawed at 37˚C before incubation on a tube rotator for 30 minutes at 4˚C, and centrifugation (13,000g, 4˚C, 15 minutes) to pellet debris. 30 μL of "input" was saved and frozen at −80˚C. 20 μL of MagStrep 'type3' beads (IBA Lifesciences) were equilibrated twice with 1 mL of wash buffer (IP buffer supplemented with 0.05% NP40) and incubated with 0.5 ml of lysate for two hours at 4˚C on a tube rotator. Tubes were placed on a magnetic rack, and beads were washed three times with 1 ml of wash buffer, and samples were shortly vortexed between washes. Bound proteins were eluted for 30 minutes using 30 μL of BXT elution buffer (IBA Lifesciences) with constant shaking at room temperature. Tubes were placed back on the magnetic rack, and the eluate was recovered and frozen at −80˚C.

Flag-immunoprecipitation was performed the same way using Anti-FLAG M2 Magnetic Beads (Sigma-Aldrich M8823) and elution was done using 3x-Flag peptide (Sigma-Aldrich F4799) at a concentration of 3 μg/50μL in IP buffer. When performing side by side Strep-affinity purification and Flag-immunoprecipitation, the same frozen cell sample was divided in two after lysis.

## Western blot

Western blot was performed using standard protocols. Briefly, protein lysate was mixed with 4x Laemmli sample buffer containing DTT and boiled for 5 minutes at 95˚C. Proteins were separated on a precast 4–20% gradient gel (Biorad, USA) and transferred on a nitrocellulose membrane using a semi-dry Trans-Blot Turbo Transfer System and Trans-Blot Turbo Transfer Buffer (Biorad, USA). Membranes were blocked for 1 hour with 5% milk in TBST (Tris-buffered saline and Tween 20) buffer, rinsed, and incubated overnight at 4˚C with primary antibodies in 2% BSA in TBST. Membranes were washed three times with TBST and incubated

for 2 hours at room temperature with horseradish peroxidase–linked secondary antibody. The chemiluminescent signal was revealed with SuperSignal West Pico PLUS Substrate (Thermo-Fisher, USA) and imaged with an Azure 600 Imaging system (Azure Biosystem, USA). Antibodies are listed in S3 Data.

### Cellular thermal shift assay (CETSA)

CETSA was performed as described [66]. Shortly, HEK-293T cells in six wells were transfected with plasmids expressing Nsp14-strep and/or SIRT5. After 48 hours, cells were harvested, washed with PBS, and resuspended in PBS supplemented with EDTA-free complete protease inhibitor cocktail (Roche). Intact cells were divided into 100-μl aliquots and heated individually at different temperatures for 3 minutes in a PCR machine (Biorad), followed by cooling for 2 minutes at room temperature. Cell suspensions were freeze-thawed three times with liquid nitrogen, and the soluble fraction was separated from the cell debris by centrifugation at $20,000 \times g$ for 20 minutes at 4˚C. Supernatants containing soluble proteins were transferred to new microcentrifuge tubes and analyzed by western blot.

### Immunofluorescence

A549 cells plated in an eight-well chamber slide (Nunc Lab-Tek II, Thermo Fisher) were transfected with 500 ng of nsp14-strep plasmid encoding nsp14-strep using Lipofectamine 3000 (Thermo Fisher). The next day, cells were fixed in PBS-buffered 4% formaldehyde at room temperature. After 15 minutes, cells were briefly rinsed with PBS once and incubated in permeabilization buffer (0.1% Triton X-100 in PBS). After 15 additional minutes, cells were incubated in blocking buffer (permeabilization buffer supplemented with 1% BSA), and further incubated in blocking buffer containing anti-Strep mouse antibody (1:1000 dilution), and anti-SIRT5 rabbit antibody (1:1000 dilution). The next day, the cells were washed with PBS three times and incubated in the blocking buffer containing anti-mouse IgG donkey antibody conjugated with Alexa 488 (1:500 dilution, Thermo Fisher), anti-rabbit IgG donkey antibody conjugated with Alexa 555 (1:500 dilution, Thermo Fisher), and for counter-staining, DAPI (1 μg/ml, Sigma Aldrich) and Phalloidin conjugated with Alexa 647 (1: 1000 dilution, Abcam). After 30 minutes, the cells were washed with PBS three times, and mounted in prolong gold anti-fade (Thermo Fisher), followed by curing overnight. The cells were visualized using a confocal fluorescence microscope (LSM 700, Carl Zeiss) at 63 X magnification, imaged, and analyzed using ZEN imaging software (blue edition, Ver 3.4, Carl Zeiss). Antibodies used are given in S3 Data.

### Protein purification and enzymatic assays

Nsp10 and Nsp14 proteins from the Wuhan strain of SARS-CoV-2 (NC_045512.2) were codon-optimized, ordered as Gblocks (IDT), and cloned into a pVFT1S expression vector using a HiFi DNA Assembly kit (NEB). Both nsp10 and nsp14 contained an N-terminal 6x-His tag, followed by a TEV cleavage site. *E. coli* BL21*(DE3) cells (Invitrogen, USA) were transformed with the nsp10 and nsp14 expression vectors and grown in LB medium containing kanamycin. Cells were induced at an $OD_{600}$ of ~0.8 with 0.5 mM IPTG for 16 hours at 16˚C. Nsp10 pellets were stored at -20˚C, and nsp14 pellets were flash-frozen with liquid nitrogen and stored at -80˚C until use. For nsp10/14 copurification, nsp10 pellets from 1 L of cells and nsp14 pellets from 6 L of cells were resuspended in lysis buffer (50 mM HEPES, pH 7.5, 300 mM NaCl, 5 mM $MgSO_4$, 30 mM imidazole, and 1% NP-40) and combined. For nsp14 purification, pellets were resuspended in lysis buffer. The pellets were lysed using sonication and clarified using centrifugation at 14,500 rpm for 40 minutes at 4˚C. The supernatant was loaded onto a HisTrap HP column (GE Healthcare, USA). Proteins were purified by fast

protein liquid chromatography and washed using two column volumes of Ni Buffer A (50 mM HEPES, pH 7.5, 300 mM NaCl, 5 mM MgSO$_4$, and 30 mM imidazole). For nsp10/14 copurification only, an additional wash was done over five column volumes using a buffer of 50 mM HEPES, pH 7.5, 300 mM NaCl, 5 mM MgSO$_4$, and 60 mM imidazole. Proteins were eluted using 50 mM HEPES, pH 7.5, 300 mM NaCl, 5 mM MgSO$_4$, and 300 mM imidazole. The elution was then concentrated and purified further using a Superdex 200 column (GE) and a buffer of 10 mM HEPES, pH 7.5, 150 mM NaCl, and 10% glycerol. The purified protein was then concentrated, flash-frozen using liquid nitrogen, and stored at -80°C.

SIRT5 purified protein was obtained commercially (BPS Bioscience, USA, Cat. 50016). In vitro desuccinylation assays were performed using Fluorogenic SIRT5 Assay Kit (BPS Bioscience, USA, Cat. 50085), following the manufacturer's instructions. Methyltransferase assays were performed using MTase-Gl Methyltransferase Assay (Promega, USA), following the manufacturer's instructions. Nsp14 and SIRT5 recombinant proteins were first incubated together in the reaction buffer, with a ratio of 1:1 corresponding to 100 nM of each protein. Desuccinylation assays were performed in a reaction buffer (25 mM Tris/HCl, pH 8.0, 137 mM NaCl, 2.7 mM KCl, 1 mM MgCl$_2$, and 0.1 mg/ml BSA) with 0.5 mM NAD+ for 30 minutes at 37°C. Methyltransferase assays were performed in reaction buffer (50 mM Hepes, pH 7.0, 6 mM KCl, 2 mM DTT, 1 mM MgCl$_2$, and 0.1 mg/ml BSA) in presence of 0.1 mM NAD + and 10 μM SAM.

## Sample preparation for proteomic analysis

HEK-293T *SIRT5-KD* cells were transfected with plasmids expressing Nsp14-strep in the presence or absence of SIRT5 with 3 biological replicates for each condition. Nsp14-strep and bound proteins were purified by affinity purification as described above, and eluted in 35 μL of elution buffer (100 mM Tris pH 7.4; 150 mM NaCl; 1 mM EDTA; 50 mM biotin). Each sample was subjected to a lysis buffer containing 5% SDS and 50 mM triethylammonium bicarbonate (TEAB) for proteomics sample preparation.

The samples enriched for NSP14 were reduced with 20 mM dithiothreitol (DTT) in 50 mM TEAB buffer at 50°C for 10 minutes, left to cool at room temperature for 10 minutes, and alkylated with 40 mM iodoacetamide (IAA) in 50 mM TEAB buffer in the dark at room temperature for 30 minutes. Samples were acidified with a final concentration of 1.2% phosphoric acid. Subsequently, 90% methanol in 100 mM TEAB was added. The entire sample volume was spun through the micro S-Trap columns (Protifi) to bind the proteins to the S-Trap column. The S-Trap columns were washed again with 90% methanol in 100 mM TEAB. S-Trap columns were placed in a clean elution tube and incubated with trypsin digestion buffer (50 mM TEAB, pH ~8) at a 1:25 ratio (protease:protein, wt:wt) for 1 hour at 47°C. The same volume of trypsin digestion buffer was added again for an overnight incubation at 37°C. Peptides were eluted from the S-Trap column first with 50 mM TEAB spun through at 1,000 x g, then with 50 mM TEAB and 0.5% formic acid at 1,000 x g, and finally with 50% acetonitrile in 0.5% formic acid at 4,000 x g. These pooled elutions were dried in a vacuum concentrator and then re-suspended in 0.2% formic acid. The re-suspended peptide samples were desalted with stage tips generated in-house using C18 disks. They were subsequently dried again in a vacuum concentrator, and re-suspended in aqueous 0.2% formic acid containing "Hyper Reaction Monitoring" indexed retention time peptide standards (iRT, Biognosys).

## Mass spectrometry analysis

Briefly, samples were analyzed by reverse-phase HPLC-ESI-MS/MS using an Eksigent Ultra Plus nano-LC 2D HPLC system (Dublin, CA) with a cHiPLC system (Eksigent) which was

directly connected to a quadrupole time-of-flight (QqTOF) TripleTOF 6600 mass spectrometer (SCIEX, Concord, CAN). After injection, peptide mixtures were loaded onto a C18 pre-column chip (200 μm x 0.4 mm ChromXP C18-CL chip, 3 μm, 120 Å, SCIEX) and washed at 2 μl/min for 10 min with the loading solvent ($H_2O$/0.1% formic acid) for desalting. Subsequently, peptides were transferred to the 75 μm x 15 cm ChromXP C18-CL chip, 3 μm, 120 Å, (SCIEX), and eluted at a flow rate of 300 nL/min with a 3 h gradient using aqueous and acetonitrile solvent buffers.

Data-dependent acquisitions: For peptide and protein identifications the mass spectrometer was operated in data-dependent acquisition (DDA) mode, where the 30 most abundant precursor ions from the survey MS1 scan (250 msec) were isolated at 1 m/z resolution for collision-induced dissociation tandem mass spectrometry (CID-MS/MS, 100 msec per MS/MS, 'high sensitivity' product ion scan mode) using the Analyst 1.7 (build 96) software with a total cycle time of 3.3 sec as previously described [67]

Data Processing: Mass spectrometric data-dependent acquisitions (DDA) were analyzed using the database search engine ProteinPilot (SCIEX 5.0 revision 4769) allowing for biological modifications and with 'emphasis' on succinylation. A fasta file was generated appending the viral NSP14 protein sequence to the human proteome fasta file. A confidence score threshold of 99 was set to filter for high quality peptide identifications. Identified protein and peptide results are provided in S1 Data.

## SARS-CoV-2 virus culture and infections

SARS-CoV-2 isolate USA-WA1/2020 (BEI NR-52281) was used for all infection studies. All live virus experiments were performed in a Biosafety Level 3 laboratory. SARS-CoV-2 stocks were propagated in Vero-E6 cells, and their sequences were verified by next-generation sequencing. Viral stock titer was measured by plaque assays.

For infection experiments, A549-ACE2 or Calu3 cells were seeded into 12- or 24-well plates and rested for at least 24 hours prior to infection. At the time of infection, medium containing compound and/or viral inoculum (MOI 0.01 or 0.1) was added to the cells. After 3 days, the supernatant was collected and mixed with two volumes of RNA STAT-60 extraction buffer (AMSbio, UK). Cells were similarly harvested by adding RNA STAT-60 extraction buffer. Samples were stored at -80˚C.

Infections of HCT-8 cells with HCoV-OC43 were performed similarly.

## Plaque assays

Viral inoculations were harvested from experiments and serially diluted in DMEM (Corning). At the time of infection, the media on Vero-E6 cells were replaced with viral inoculation for 1 hour. After the 1-hour absorption period, 2.5% Avicel (Dupont, RC-591 was layered on the cells and incubated for 72 hours. Then, Avicel was removed and cells were fixed in 10% formalin for 1 hour, stained with crystal violet for 10 minutes, and washed multiple times with water. Plaques were counted and averaged from two technical replicates.

## RNA extraction and RT-qPCR

RNA in cells and supernatants resuspended in RNA STAT-60 buffer was extracted using Direct-zol RNA Miniprep kit (ZymoResearch, USA). For extraction from the supernatant, RNA was eluted in 20 μL of water, and 18 μL was directly used for reverse-transcription. RNA extracted from cells was DNAse-treated, eluted in 30 μL quantified by Nanodrop, and 2 μg of RNA was used reverse-transcription. Reverse-transcription was performed using the High-Capacity cDNA Reverse Transcription Kit (ThermoFisher), and quantitative PCR was done

using a BioRad qPCR machine and Sybr Green (ThermoFisher). RT-qPCRs were normalized using the ΔΔCt method with the reference genes *ACTIN* and *GAPDH*. qPCR primers are given in S3 Data. Missing data points in RT-qPCR figures represent samples where RNA extraction was of poor quality. No outliers were removed.

### Statistics and reproducibility

Experiments were carried out in multiple replicates. For affinity-purification and western blot data, one representative experiment out of several replicates is shown. For statistical analysis of RT-qPCR data, we used ordinary one-way ANOVA, followed by Holm–Šidák multiple comparisons test, with a single pooled variance. For RT-qPCR in Fig 7, we used unpaired two-tailed t-tests. Plaque assay data do not satisfy the normality condition required for parametric tests, but are closer to a lognormal distribution [68]. As a consequence, statistical tests on plaque assay data were performed on log-transformed data. Analyses were run using GraphPad Prism version 9.1.2 for macOS (GraphPad Software, USA, www.graphpad.com). Exact p-values and summaries are reported in the text and figures, respectively.

### RNA-sequencing preparation and analysis

Library preparation and sequencing were performed by the DNA Technologies and Expression Analysis Core at the UC Davis Genome Center (Davis, CA, USA). Strand-specific and barcode-indexed RNA-seq libraries were generated from 500 ng of total RNA each, after poly-A enrichment, using the Kapa mRNA-seq Hyper kit (Kapa Biosystems-Roche, Basel, Switzerland), following the instructions of the manufacturer. The fragment size distribution of the libraries was verified via micro-capillary gel electrophoresis on a LabChip GX system (PerkinElmer, Waltham, MA). The libraries were quantified by fluorometry on a Qubit instrument (LifeTechnologies, Carlsbad, CA) and pooled in equimolar ratios. The pool was quantified by qPCR with a Kapa Library Quant kit (Kapa Biosystems) and sequenced on an Illumina NovaSeq 6000 (Illumina, San Diego, CA) with paired-end 150-bp reads.

Paired-end sequencing reads were mapped to a composite human/SARS-CoV-2 genome using Subread Aligner v2.0.3 [69]. A genome index was constructed using GRCh38 genome build with Gencode v38 annotation of the transcriptome, and Genbank MT246667.1 for the sequence of SARS-CoV-2, USA/WA-1/2020 isolate. Reads mapping to annotated genes were counted using Subread featurecount v2.0.3 [70]. Downstream analyses were performed with R. Differential gene expression analysis was done with DEseq2, which was also used to generate normalized gene counts [71]. Low-expressed genes with less than three counts in at least three out of 16 samples were excluded from downstream analysis. q-Values were calculated using the q-value R package v2.24.0. For DEseq2 analysis, we used a one-factor design with four groups (WT mock, KO mock, WT infected and KO infected) and then likelihood ratio testing (LRT) to find all genes that were differentially expressed between at least two groups (q-value threshold <0.01), and with a basemean expression >15. Consensus clustering of the 3221 differentially expressed genes was performed with the degPatterns function of R DEgreport package v1.28.0, using default parameters and rlog-transformed counts. This generated eight clusters. Over-representation of biological gene sets in the gene clusters was investigated using the R clusterProfiler package and enricher function [72]. Gene sets were downloaded from the MSIG data bank via the msigdbr R-project package, including "Hallmark," and "Reactome". [73]. Gene sets were considered significantly enriched in a cluster if q values were < 0.05. For analysis of restriction factors in Fig 6, we first selected genes in clusters 1 to 8 that belonged to the hallmark curated data set "Interferon Alpha response". We complemented this list with additional genes from clusters 7 and 8 that belonged to the "Interferon Gamma

response", "Inflammatory response" and "TNFa signaling via NfKB" hallmark datasets, and finally added genes that we manually identified as potential restriction factors from literature searches. Average log2 fold-change compared to mock-infected WT was calculated and plotted, as well as the q-value between mock-infected WT and KO. Code developed for this study is available at https://github.com/mariuswalter/SIRT5_paper. This analysis relied heavily on code made available by the laboratory of Denis Goldfarb (https://github.com/GoldfarbLab/H522_paper_figures), and described in ref [45].

## Supporting information

**S1 Fig. Characterization of inhibitors. A**. Western Blot after transfection of HEK-293T cells with SIRT5 WT and catalytic mutants, with or without treatment for 6h with the proteasome inhibitor MG-132. No major defect in protein folding could be detected. **B**. SIRT5 in vitro desuccinylation activity in the presence of Sirt5-i inhibitor. n = 3. **C**. Diagram of the NAD salvage pathway. Inhibition of NAMPT by FK866 inhibitor depletes cellular NMN and NAD levels. Supplementation by exogenous NMN rescues NAD.
(TIF)

**S2 Fig. Levels of viral restriction factors. A**. Principal component analysis of RNA-seq samples, showing that replicates are well separated based on knockout and infection status. **B**. Normalized gene count of interferon-stimulated genes and restriction factors, from Fig 7.
(TIF)

**S3 Fig. Role of SIRT5 in the RIG-1/MAVS antiviral signaling pathway. A**. Recognition of cytosolic viral RNA by RIG-1 or MDA5 leads to MAVS aggregation on the mitochondrial surface, which in turn causes type I interferon production. SIRT5 desuccinylation of MAVS impairs MAVS aggregation and reduces interferon production. Adapted from Liu et al. [30].
(TIF)

**S1 Data. Mass Spectrometry data.** Proteins and peptides identified by mass spectrometry after affinity purification of Nsp14 in HEK293 cells. Two conditions were analyzed: Nsp14 transfected alone, or Nsp14 co-transfected with SIRT5.
(XLSX)

**S2 Data. RNA-seq data.** RNA-seq results of WT and *SIRT5*-KO A549-ACE2 cells infected or mock-infected with SARS-CoV-2 for 3 days at MOI = 0.1. First tab show normalized counts and log2 fold change (l2fc) between the different condition. Tabs 2–4 show results of gene ontology analysis between the different conditions.
(XLSX)

**S3 Data. Reagents.** CRISPR gRNA sequences, primer sequences, antibodies, and small molecule inhibitors used in the study.
(PDF)

## Acknowledgments

We thank the QCRG Virology group at UCSF for technical and conceptual help, as well as members of the Verdin and Ott lab, in particular Rebeccah Riley, Rosalba Perrone and Anthony Covarrubias. We thank Michelle Moritz at UCSF for attempting co-purification experiments in *E. coli*. We thank David Gordon for quickly sharing the plasmid library early in the study and Max Greenberg at Paris University for sharing the Transposase plasmid.

## Author Contributions

**Conceptualization:** Marius Walter, Melanie Ott, Eric Verdin.

**Data curation:** Marius Walter, Birgit Schilling.

**Formal analysis:** Marius Walter.

**Funding acquisition:** Melanie Ott, Eric Verdin.

**Investigation:** Marius Walter, Irene P. Chen, Albert Vallejo-Gracia, Ik-Jung Kim, Olga Bielska, Victor L. Lam, Jennifer M. Hayashi, Andrew Cruz, Samah Shah, Frank W. Soveg, Birgit Schilling.

**Methodology:** Marius Walter, Irene P. Chen, Birgit Schilling.

**Resources:** John D. Gross, Nevan J. Krogan, Keith R. Jerome.

**Supervision:** John D. Gross, Birgit Schilling, Melanie Ott, Eric Verdin.

**Validation:** Irene P. Chen.

**Writing – original draft:** Marius Walter, Eric Verdin.

**Writing – review & editing:** Marius Walter, Melanie Ott, Eric Verdin.

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
