## [Decision Letter · Decision Letter 0]

9 Mar 2022

Dear %TITLE% Walter,

Thank you very much for submitting your manuscript "SIRT5 is a proviral factor that interacts with SARS-CoV-2 Nsp14 protein" for consideration at PLOS Pathogens. As with all papers reviewed by the journal, your manuscript was reviewed by members of the editorial board and by several independent reviewers. In light of the reviews (below this email), we would like to invite the resubmission of a significantly-revised version that takes into account the reviewers' comments.

While the overall consensus of reviewers was positive, they felt that some additional experiments are required to make this manuscript suitable for publication in PLOS Pathogens.

From the multiple experiments suggested by reviewers I am guiding the authors towards the following experiments for a revised manuscript:

1) To strengthen evidence for the proposed link between NSP14/SIRT5 and innate immune activation: This seems to be the major hurdle to overcome. I suggest following reviewer 1 suggestions in investigating in mechanistic detail the complex's action on factors of the innate immune cascade, i.e. MAVS. The authors should also examine whether expression of Nsp14 would induce changes in innate response similar to a loss or induction of SIRT5. The methodology of generating recombinant SARS-CoV-2 may not be open to the authors in order to follow through with modulating NSP14 binding to SIRT5 and probing for the impact on innate immune activation. As an alternative, the authors may utilize cellular tools to modulate innate activation, i.e. use of IFN-competent and -incompetent cells, or Jak/Stat inhibitors, and investigate whether the phenotype holds. Finally, SIRT5 KO cells complemented with a mutant of SIRT5 that lacks binding to Nsp14 could provide some insights.

2) To strengthen the physiological relevance of findings: Perform synergy experiments with combined SIRT1/5 inhibitors and/or SIRT5 inhibitors with remdesivir or other available antiviral drugs.

3) To strengthen mechanistic detail of SIRT5/NSP14 interaction: The path forward to biochemical studies seem to be barred by difficulties with protein purification. I suggest incorporating reviewer 3's comments by adding predicted 3D models of the different mutants, and otherwise tackling the reviewer's points in the in discussion section.

This editor recognizes that mouse experiments to perform suggested pathogenicity studies may be outside of the scope of this manuscript.

We cannot make any decision about publication until we have seen the revised manuscript and your response to the reviewers' comments. Your revised manuscript is also likely to be sent to reviewers for further evaluation.

Sincerely,

Meike Dittmann, Ph.D.

Associate Editor

PLOS Pathogens

Andrew Pekosz

Section Editor

PLOS Pathogens

Kasturi Haldar

Editor-in-Chief

PLOS Pathogens

orcid.org/0000-0001-5065-158X

Michael Malim

Editor-in-Chief

PLOS Pathogens

orcid.org/0000-0002-7699-2064

While the overall consensus of reviewers was positive, they felt that some additional experiments are required to make this manuscript suitable for publication in PLOS Pathogens.

From the multiple experiments suggested by reviewers I am guiding the authors towards the following experiments for a revised manuscript:

1) To strengthen evidence for the proposed link between NSP14/SIRT5 and innate immune activation: This seems to be the major hurdle to overcome. I suggest following reviewer 1 suggestions in investigating in mechanistic detail the complex's action on factors of the innate immune cascade, i.e. MAVS. The authors should also examine whether expression of Nsp14 would induce changes in innate response similar to a loss or induction of SIRT5. The methodology of generating recombinant SARS-CoV-2 may not be open to the authors in order to follow through with modulating NSP14 binding to SIRT5 and probing for the impact on innate immune activation. As an alternative, the authors may utilize cellular tools to modulate innate activation, i.e. use of IFN-competent and -incompetent cells, or Jak/Stat inhibitors, and investigate whether the phenotype holds. Finally, SIRT5 KO cells complemented with a mutant of SIRT5 that lacks binding to Nsp14 could provide some insights.

2) To strengthen the physiological relevance of findings: Perform synergy experiments with combined SIRT1/5 inhibitors and/or SIRT5 inhibitors with remdesivir or other available antiviral drugs.

3) To strengthen mechanistic detail of SIRT5/NSP14 interaction: The path forward to biochemical studies seem to be barred by difficulties with protein purification. I suggest incorporating reviewer 3's comments by adding predicted 3D models of the different mutants, and otherwise tackling the reviewer's points in the in discussion section.

This editor recognizes that mouse experiments to perform suggested pathogenicity studies may be outside of the scope of this manuscript.

Reviewer's Responses to Questions

**Part I - Summary**

Reviewer #1: This manuscript by Walter, Verdin and colleagues examines the effects of the sirtuin SIRT5 in cellular models of SARS-CoV-2 infection. SIRT5 is a member of the sirtuin deacylase family; SIRT5 has been shown, in work by the Verdin group and others, to act as a cellular desuccinylase, demalonylase, and deglutarylase, rather than as a deacetylase. The authors find that SIRT5 interacts with Nsp14, a viral protein with multiple enzymatic activities, in a SIRT5-activity dependent manner. The authors further show that genetic or pharmacologic inhibition of SIRT5 results in modest decreases in viral mRNA levels and viral titer in cultured cells infected with SARS-CoV-2. Via transcriptomic studies, the authors show that SIRT5 impacts multiple pathways in cells, including autophagy and innate immunity.

It goes without saying that Covid-19 remains an enormous global public health challenge. Thus, identifying new players in SARS-CoV-2 pathobiology, such as SIRT5, is of great relevance. With that said, there are issues with this manuscript that need to be addressed before it is acceptable for publication, in my view.

Reviewer #2: Viruses, including the pandemic SARS-CoV-2, need to manipulate the cellular environment to facilitate their own replication and propagation. To this end, they engage cellular factors and exploit them for their purposes. These pro-viral factors are often targeted and hijacked by viral proteins.

In this manuscript, Walter and colleagues identified the cellular deacylase/desuccinylase/demalonylase SIRT5 (and SIRT1) as an interaction partner of the SARS-CoV-2 non-structural protein 14. While the interaction depends on the activity/active domain of SIRT5, it does not remove PTMs from Nsp14. The authors propose that by targeting SIRT5, SARS-CoV-2 reduces innate immune activation, as cells lacking SIRT5 show increased levels of known ISGs that antagonise SARS-CoV-2.

Overall the data presented in the manuscript is already quite convincing, generally supports the main conclusion and certainly provides an interesting and novel description of a (non-methylase) function of Nsp14 and identification of its cellular interaction partner SIRT5 as a pro-viral protein. However, the link between Nsp14/SIRT5 and the innate immune induction phenotype is a bit weak, as well as the impact of the Nsp14-SIRT5 interaction on virus replication.

I have only a few suggestions to improve the current manuscript.

Reviewer #3: Walter et al. demonstrates that SIRT5 (and to some extent SIRT1) facilitates SARS-CoV-2 replication in vitro. Both proteins stably interact with Nsp14, and the catalytic activity of SIRT5 and SIRT1 are necessary for this interaction. Based on their results, they propose SIRT5 and SIRT1 to be proviral factors and could be a new therapeutic targets for Covid19 treatment.

The authors show compelling evidence that SIRT5 binds Nsp14 using immunoprecipitations from cell lines. They also prove that SIRT5 catalytic activity is needed for this interaction and that SIRT5/SIRT1 KD or inhibition lowers viral propagation in cell lines. Experiments are performed at high quality and the manuscript is well written, however the manuscript presented here has some key limitations.

**Part II – Major Issues: Key Experiments Required for Acceptance**

Reviewer #1: 1. The authors show that SIRT5 interacts with Nsp14, but also that SIRT5-inhibited cells show an enhanced innate immune activation state that actually preexists infection. Do the authors have any evidence that the SIRT5-Nsp14 interaction actually has any role in modulating the outcome of SARS-CoV-2 infection? It seems to me that this interaction may be irrelevant to the role of SIRT5 in promoting viral pathogenesis identified by the authors, which could be driven exclusively by the effects of SIRT5 genotype on innate immunity.

2. The effects of SIRT5 deficiency/inhibition on viral titers seem quite modest, mostly less than a log-fold. Thus, the physiologic relevance of these findings remains somewhat uncertain. Since the authors also identify a role for SIRT1 inhibition in modulating cellular SARS-CoV-2 responses, perhaps it is worth testing the effects of combined SIRT1/SIRT5 inhibition, or the effects of combing SIRT5 inhibition together with known effective anti-viral therapy (e.g. remdesivir, Paxlovid, and the like).

3. The manuscript would be much stronger if the authors had evaluated SIRT5 KO mice for their response to SARS-CoV-2 infection, and this might address point 2. Such animals might well show improved responses over those observed in cell culture, since the totality of their immune responses might well be enhanced (not only through cell-autonomous effects in the cells themselves). I fully recognize the logistical difficulties involved in such studies, which may be outside the scope of this manuscript.

Reviewer #2: Major issues:

While the authors check in detail whether SIRT5 regulates PTMs on Nsp14, and find no evidence of that they should check whether binding of Nsp14 to SIRT15 alters its impact on other cellular factors, especially in the IFN cascade, e.g. MAVS (K7 succinylation). Furthermore, the authors should examine whether expression of Nsp14 would induce changes in innate response similar to a loss or induction of SIRT5. This may provide a link between Nsp14 and innate immune defences. Ideally, a recombinant SARS-CoV-2 virus lacking binding of Nsp14 to SIRT5 would be more immunostimulatory and attenuated in immunocompetent cells. Alternatively, SIRT5 KO cells complemented with a mutant of SIRT5 that lacks binding to Nsp14 could provide some insights (ideally with intact catalytic activity!).

It is a bit unusual that catalytic activity necessary for binding. As the in vitro assays suggest it is a direct binding, can this be outcompeted in vitro by a SIRT5 inhibitor?

Reviewer #3: Key limitations:

(i) The authors were not able to reconstitute the SIRT5-Nsp14 complex in vitro using recombinant proteins purified from E. coli. This means that the authors are probably missing key components of the complex as these two proteins might not be binding each other directly. This would be a key mechanistic point to try to tackle (for example with mass spectrometry identification of SIRT5-Nsp14 complex components).

(ii) The authors show that Nsp14 from HCoV-OC43 does not bind SIRT5. They also show that SIRT5 catalytic inhibition also lowers HCoV-OC43 replication efficiency. This suggests that SIRT5 might be exerting its antiviral properties completely independently from binding Nsp14.

Please see my specific comments below:

Major points:

1.) Figure 2B shows the binding between Nsp14 and SIRT5 catalytic mutants. Some mutations completely abolish binding while others have no or only mild effect. It is not evident if the mutations (and the Sirt5-i) affect the folding of SIRT5 and that causes the reduction in binding and enzymatic activity. The authors should establish experimentally the effect of these mutations on SIRT5 enzymatic activity and the folding of the protein. Since mutation Y102F maintains the binding to Nsp14, this could give a perfect tool to de-couple catalytic activity of SIRT5 from Nsp14 binding. If SIRT5 Y102F mutant can rescue the effect of SIRT5 KD on viral replication the authors could answer a key question if SIRT5-Nsp14 binding or just SIRT5 enzymatic activity is needed for proper viral replication.

2.) In Figure 3D and E the authors claim that SRT1720 and Resveratrol do not have an effect on Nsp14 and SIRT1 binding. Based on their blots, there is a clear reduction in binding between Nsp14 and SIRT1 when SRT1720 is used at 25, 50 uM and Resveratrol used at 25 and 100 uM.

3.) Unfortunately, the authors were not able to uncover the molecular functions of the Nsp14/SIRT5 complex. In Figure 3G the authors only show the effect of SIRT5 on the methyltransferase activity of Nsp14 (does not affect it). They state that since Nsp10 is needed for the ExoN activity of Nsp14 and that SIRT5 and Nsp10 are in separate complexes they did not evaluate the potential effect of SIRT5 on the ExoN activity of Nsp14. While this is true, the authors should have considered the following hypothesis: They have proved that Nsp10 and SIRT5 is competing to bind Nsp14. Therefore, SIRT5 might inhibit Nsp14 ExoN activity during viral replication by outcompeting Nsp10, which has been shown to be vital for SARS-CoV-2 replication.

**Part III – Minor Issues: Editorial and Data Presentation Modifications**

Reviewer #1: 4. (Minor): in their Nsp14 IP/IBs, the authors should examine lysine glutarylation, in addition to the marks examined.

Reviewer #2: Minor issues:

A double KO of SIRT1 or SIRT5 should be added, to show whether there are two independent functions of these protein targeted, or whether it is additive effects.

I strongly recommend to improve the figure legends, i.e. show number of repeats, statistics used, antibodies used etc

The western blots would benefit from black boxes around them, currently the fade into the white paper background too much as the contrast is very high.

The phrase ‘stably interact’ sounds a bit odd to me and is used multiple times throughput the manuscript. What do the authors mean by this?

Fig. 1 C: Please add a quantification of the colocalization

Positive controls for e.g. a SIRT5-dependent deacetylation missing in Fig. 2E.

Fig. 2F: Can the authors state whether putative PTM sites were covered by the peptides in the MS?

Fig. 3A: Please include SIRT5 as a control and show the IP blot of Strep.

Reviewer #3: Minor points:

1.) Figure 5 shows the up and down regulated genes in the SIRT5 WT vs. KO cells w/wo viral infection. The authors should discuss further how many of the genes with differential expression have the potential to restrict viral replication in an in vitro environment without a functional innate immune system present in their assays.

PLOS authors have the option to publish the peer review history of their article (what does this mean?). If published, this will include your full peer review and any attached files.

Reviewer #1: No

Reviewer #2: No

Reviewer #3: No
---

## [Decision Letter · Decision Letter 1]

18 Aug 2022

Dear Walter,

We are pleased to inform you that your manuscript 'SIRT5 is a proviral factor that interacts with SARS-CoV-2 Nsp14 protein' has been provisionally accepted for publication in PLOS Pathogens.

Best regards,

Meike Dittmann, Ph.D.

Associate Editor

PLOS Pathogens

Andrew Pekosz

Section Editor

PLOS Pathogens

Kasturi Haldar

Editor-in-Chief

PLOS Pathogens

orcid.org/0000-0001-5065-158X

Michael Malim

Editor-in-Chief

PLOS Pathogens

orcid.org/0000-0002-7699-2064

Reviewer Comments (if any, and for reference):

Reviewer's Responses to Questions

**Part I - Summary**

Reviewer #2: The authors have significantly strengthend the manuscript and addressed my concerns. I also appreciate the amount of work that went into addressing all reviewer's suggestions, despite the data not being included in the final revision.

Reviewer #3: I appreciate the honesty and effort of the authors to try to address all my concerns. The manuscript is now improved. The results with rescuing the SIRT5 KO cells with WT or the H158Y or the Y102F mutant SIRT5 is indeed challenging to interpret. The authors proposed that this might be due to an artifact caused by high protein expression levels, which might be possible.

**Part II – Major Issues: Key Experiments Required for Acceptance**

Reviewer #2: (No Response)

Reviewer #3: (No Response)

**Part III – Minor Issues: Editorial and Data Presentation Modifications**

Reviewer #2: (No Response)

Reviewer #3: (No Response)

PLOS authors have the option to publish the peer review history of their article (what does this mean?). If published, this will include your full peer review and any attached files.

Reviewer #2: No

Reviewer #3: No

---

## [Editor Report · Acceptance letter]

6 Sep 2022

Dear Dr. Walter,

We are delighted to inform you that your manuscript, "SIRT5 is a proviral factor that interacts with SARS-CoV-2 Nsp14 protein," has been formally accepted for publication in PLOS Pathogens.

Best regards,

Kasturi Haldar

Editor-in-Chief

PLOS Pathogens

orcid.org/0000-0001-5065-158X

Michael Malim

Editor-in-Chief

PLOS Pathogens

orcid.org/0000-0002-7699-2064